# Preference-driven Knowledge Distillation for Few-shot Node Classification

**Xing Wei**[1]  **Chunchun Chen**[2]  **Rui Fan**[1,2,3]  **Xiaofeng Cao**[4]  **Sourav Medya**[5]  **Wei Ye**[1,2*]

[1] College of Electronic and Information Engineering, Tongji University, China
[2] Shanghai Research Institute for Intelligent Autonomous Systems, Tongji University, China
[3] National Key Laboratory of Human-Machine Hybrid Augmented Intelligence,
Xi'an Jiaotong University, China
[4] School of Computer Science and Technology, Tongji University, China
[5] Department of Computer Science, University of Illinois Chicago, USA
{xing627, c2chen, yew}@tongji.edu.cn, rui.fan@ieee.org,
xiaofeng.cao.uts@gmail.com, medya@uic.edu

## Abstract

Graph neural networks (GNNs) can efficiently process text-attributed graphs (TAGs) due to their message-passing mechanisms, but their training heavily relies on the human-annotated labels. Moreover, the complex and diverse local topologies of nodes of real-world TAGs make it challenging for a single mechanism to handle. Large language models (LLMs) perform well in zero-/few-shot learning on TAGs but suffer from a scalability challenge. Therefore, we propose a preference-driven knowledge distillation (PKD) framework to synergize the complementary strengths of LLMs and various GNNs for few-shot node classification. Specifically, we develop a GNN-preference-driven node selector that effectively promotes prediction distillation from LLMs to teacher GNNs. To further tackle nodes' intricate local topologies, we develop a node-preference-driven GNN selector that identifies the most suitable teacher GNN for each node, thereby facilitating tailored knowledge distillation from teacher GNNs to the student GNN. Extensive experiments validate the efficacy of our proposed framework in few-shot node classification on real-world TAGs. Our code is available at https://github.com/GEEX-Weixing/PKD.

## 1  Introduction

Text-attributed graphs (TAGs [1]), such as citation, webpage, and product graphs [2, 3], have nodes associated with text attributes. Graph neural networks (GNNs) [4, 5] have demonstrated excellent performance and efficiency in node classification on TAGs, which are supported by high-quality labels and effective message-passing mechanisms [6]. However, the manual labeling of nodes is undoubtedly a tedious, expensive, and time-consuming task [7, 8]. In many scenarios, only a few node labels are available. Additionally, nodes often have complex and diverse interaction relationships with each other—their local topologies are intricate—which challenge traditional GNNs with fixed message-passing mechanisms. Compared with GNNs, large language models (LLMs) exhibit impressive zero-/few-shot learning capabilities on TAGs [9, 10, 11]. But the large parameter scale considerably hinders their inference efficiency [12].

A natural idea is *to blend their complementary strengths for few-shot node classification on TAGs*. Knowledge distillation (KD) [13] is a feasible solution. However, directly distilling knowledge from the LLM to GNN is impractical. Firstly, the discrepancy of decoder-only (LLMs) and encoder-only

---

*Corresponding Author

39th Conference on Neural Information Processing Systems (NeurIPS 2025).

(GNNs) leads to fundamentally different characteristics in their embedding spaces [14]. And the huge embedding-dimension difference needs sophisticated embedding alignment and also brings high training cost [15]. In contrast, conducting prediction distillation from LLMs to GNNs by annotating node labels can efficiently alleviate the label scarcity and scalability dilemma [16]. The critical question is how to select the nodes for the LLM's label annotation to effectively enhance teacher GNNs. Generally, one may use uncertainty [17] as a selection metric in the embedding space of GNN. However, owing to nodes' diverse semantic and complex structural attributes (e.g., local topologies), a single GNN cannot capture the essences of nodes completely [18]. Therefore, we investigate the embedding spaces of various-architecture GNNs to effectively mitigate cognitive limitations [19] associated with relying on a single GNN, thereby better selecting nodes for LLM's label annotations.

Nevertheless, since nodes have intricate local topologies, which need tailored message-passing mechanisms, *how to tailor for each node the most appropriate message-passing mechanism* is another challenge. Different GNNs provide different prediction attributes for each node during the learning process [20], encompassing the understandings of its topologies, its interaction relationships to other nodes, and its latent patterns. These node-specific attribute differences suggest that a single message-passing mechanism cannot fundamentally handle the entire graph. Some studies [21, 18] distill knowledge sequentially or simultaneously from teacher GNNs without taking into account the node-specific local topologies, resulting in no obvious performance improvement or even performance degradation [22]. Therefore, it is essential to identify the GNN message-passing mechanisms that align with the node-specific attributes.

To this end, we propose a preference-driven knowledge distillation (PKD) framework that unites the complementary strengths of LLMs and various-architecture GNNs for few-shot node classification on TAGs. It mainly includes two modules: GNN-preference-driven Node Selector (GNS) and Node-preference-driven GNN Selector (NGS). The prerequisite of GNS is that the LLM should be able to comprehend the graph topology. Thus, we develop the graph topology aware (GTA) prompts to fine-tune the LLM, enhancing its capacity to comprehend graph topology. GNS fully exploits nodes' prediction discrepancies among various GNNs to decide nodes whose labels are annotated by the LLM will effectively enhance teacher GNNs, facilitating knowledge distillation from the LLM to teacher GNNs. NGS selects for each node the most appropriate GNN message-passing mechanism, facilitating the tailored knowledge distillation from various teacher GNNs to the student GNN. It regards the fine-tuned LLM as the RL-based (reinforcement learning) agent, which treats all textualized node-specific attributes (including node's semantic, structure, and prediction attributes) as state and the student GNN's performance as reward. Our contributions can be summarized as follows:

- We introduce a preference-driven knowledge distillation (PKD) framework to synergize the complementary strengths of the LLM and various GNNs ingeniously for few-shot node classification on TAGs;

- We propose a GNN-preference-driven node selector, effectively determining nodes for annotation by the LLM and promoting knowledge distillation from the LLM to teacher GNNs;

- We propose a node-preference-driven GNN selector to tailor for each node the most appropriate message-passing mechanism, promoting knowledge distillation from teacher GNNs to the student GNN;

- We validate the efficacy of PKD for few-shot node classification on nine TAGs. The experiments show that it even defeats some state-of-the-art methods that use more node labels.

## 2   Related Work

### 2.1   Graph Neural Networks

The field of graph learning has been dominated by GNNs. Early GCN [4] introduces a spectral-based graph convolution operation to propagate node information through the graph. GAT [23] uses attention mechanisms to weigh neighbors' contributions, enabling adaptive learning of neighborhood importance. APPNP [24] enhances message passing by using personalized propagation with a power iteration approach, improving label propagation on graphs. $H_2$GCN [25] extends GCNs by incorporating higher-order neighborhood information to improve representation power. GPRGNN [26]

combines graph convolution with residual connections to improve propagation efficiency, particularly in graphs with diverse node degrees. HoloNets [27] introduces a dual-filter mechanism with spectral response, extending spectral convolutions to directed graphs. DirGNN [28] defines the in-neighbors and out-neighbors and performs separate propagation and aggregation, improving the message passing through the incorporation of edge directionality. To deal with label scarcity, GCNII [29] introduces initial residual connections and identity mapping to construct a deep GNN while EGNN [30] enforces equivariance constraints for the enhancement of data efficiency and generalization. AGST [31] and IceBerg [32] leverage the different self-training [33] methods to effectively utilize unlabeled nodes.

## 2.2 Knowledge Distillation

KD is not only used for model compression, but for strengthening purposeful abilities of the student model. GFL [34] extracts structural knowledge from a pre-prepared similar auxiliary graph, distilling it to the target graph for enhancing few-shot node classification performance. KDGA [35] utilizes multiple graph augmentation strategies to make student GNN produce robust node representations after distillation. MSKD [36] mitigates the diverse classification situations requiring for different nodes by capturing multi-scale topological semantics distilled from varying layers. However, the capability of an individual teacher is inherently limited. BGNN [21] distills complementary knowledge from multiple GNN teachers sequentially and integrate it by the adaptive temperature parameter and weight boosting modules. MTAAM [22] distills knowledge of multiple teacher GNNs into an MLP-student, offering quick inference speed without compromising accuracy. FairGKD [37] obtains equitable and informative node representations by synergizing multiple GNN experts into a teacher. DMKD [18] harnesses complementary knowledge from various GNNs and conducts layer-level knowledge distillation to mitigate the constraint of a single teacher. Furthermore, [14] is a label-free method that proposes the LLM-GNN. It uses LLMs to get high-quality annotation through active and confidence-awareness node selection, thereby circumventing the difficulty of label annotation by humans. LinguGKD [15] introduces a kind of ingenious contrastive learning to align the LLM's semantic features with GNN's structural features to achieve knowledge transfer. Most of the above knowledge distillation methods do not tailor for each node the most appropriate message-passing mechanism and underperform on few-shot node classification.

## 3 Method: PKD

In this section, we present the preference-driven knowledge distillation (PKD) framework. PKD involves two key modules: GNN-preference-driven Node Selector (GNS) and Node-preference-driven GNN Selector (NGS). The main goal of the former module is to select node groups whose labels are annotated by the LLM will drastically enhance teacher GNNs. The main goal of the latter module is to select the most appropriate teacher GNN for each node, thereby tackling the complication of node-specific local topologies. The PKD framework is illustrated in detail in Figure 1.

### 3.1 Background

A text-attributed graph (TAG) is denoted by $\mathcal{G}_T = (\mathcal{V}, \mathcal{E}, \mathbf{X}, \mathbf{A}, \mathbf{T})$, where $\mathcal{V} = \{v_1, \ldots, v_N\}$ is a set of nodes with semantic attributes $\mathbf{T} = \{\mathbf{t}_1, \ldots, \mathbf{t}_N\}$ and $\mathcal{E}$ is a set of edges. Each semantic attribute can then be encoded as a sentence embedding $\mathbf{X} = [\mathbf{x}_1, \ldots, \mathbf{x}_i, \ldots, \mathbf{x}_N] \in \mathbb{R}^{N \times F}$ with the help of language models. $\mathbf{A} \in \mathbb{R}^{N \times N}$ is the adjacency matrix. Given the few-shot node classification task, let $\mathcal{D}_L = \{(\mathbf{x}_i, \mathbf{y}_i)\}_{i=1}^{Q} \ (Q \ll N)$ be the set of labeled nodes with $\mathbf{y}_i$ as the one-hot label of the training sample $\mathbf{x}_i$ and $\mathcal{D}_U$ be the set of unlabeled nodes, respectively. The goal is to accurately predict the labels of nodes that belong to $\mathcal{D}_U$ given few labeled nodes in $\mathcal{D}_L$.

We assume $B$ teacher GNNs denoted by $\{T_b\}_{b=1}^{B}$, and $f_{T_b}^{\theta}$ is the model parameters of $T_b$. The $B$ logit outputs of teacher GNNs for node $v_i$ are written as $\mathbf{z}_i^T = [\mathbf{z}_{i,1}^T, \ldots, \mathbf{z}_{i,b}^T, \ldots, \mathbf{z}_{i,B}^T]$, which is the concatenation of the logit of each teacher $\mathbf{z}_{i,b}^T = [z_{i,b,1}^T, \ldots, z_{i,b,c}^T, \ldots, z_{i,b,C}^T] \ (1 \le b \le B)$, where $z_{i,b,c}^T$ is the probability of $v_i$ belonging to class $c \ (1 \le c \le C)$ computed by teacher $T_b$. Our final objective for the KD from node-preference GNNs to the student GNN can be divided into three parts:

$$\mathcal{L}_{KD} = \alpha \cdot (-\frac{1}{N} \sum_i^N \widetilde{\mathbf{z}}_i^T \cdot \log f_S^{\theta}(\mathbf{x}_i)) + \beta \cdot (-\frac{1}{Q} \sum_i^Q \mathbf{y}_i \cdot \log f_S^{\theta}(\mathbf{x}_i)) + \gamma \cdot (\frac{1}{N} \sum_{i=1}^N H(f_S^{\theta}(\mathbf{x}_i))) \quad (1)$$

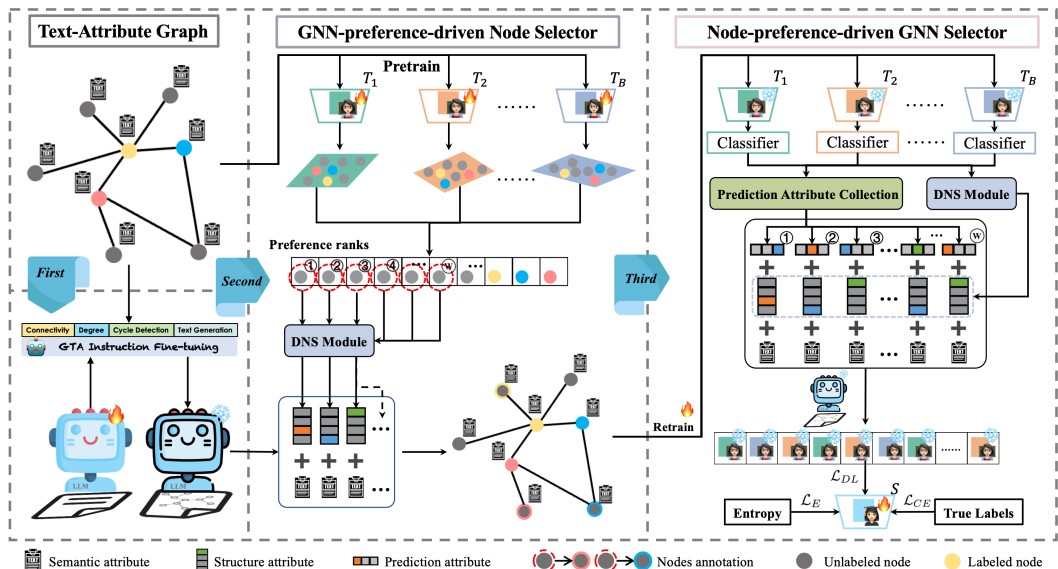

Figure 1: Overview of PKD. The framework has two key modules: GNN-preference-driven Node Selector (GNS) and Node-preference-driven GNN Selector (NGS). Before starting GNS, we first fine-tune the LLM with GTA prompts to enable it to comprehend graph properties. In the GNS module, we exploit the proposed $K$-uncertainty based on the node prediction uncertainty in each teacher GNN's embedding space to select nodes. For effectively exploiting the LLM to annotate those selected nodes, we combine the semantic attributes and structure attributes derived from the proposed Distance-based Neighbor Selector (DNS) module on these nodes to construct prompt, promoting the prediction distillation from the fine-tuned LLM to teacher GNNs ($T_1, T_2, \ldots, T_B$). In the NGS module, we select for each node the most appropriate teacher GNN for tailored knowledge distillation. The teacher GNN selection is achieved by reinforcement learning with the fine-tuned LLM as agent.

where $\alpha, \beta, \gamma$ are hyper-parameters to balance three losses. For student GNN with parameters $f_S^\theta$, the first loss, distillation loss $\mathcal{L}_{DL}$, is defined as the cross-entropy between the predictions of the teacher GNNs and that of the student GNN. The $f_S^\theta(\mathbf{x}_i)$ is the Softmax output of student GNN and it denotes the probability distribution of $v_i$ belonging to class $c$. $\widetilde{\mathbf{z}}_i^T = \mathbf{m}_i \otimes \mathbf{z}_i^T$, where $\mathbf{m}_i$ is a one-hot vector denoting which teacher GNN is preferred by $v_i$. The second loss, $\mathcal{L}_{CE}$, is the cross-entropy loss in the training of student GNN. Inspired by [38], we add $\mathcal{L}_E$ to the objective as the last part, which makes the logits of student GNN closer to one-hot vectors. The $H(\cdot)$ denotes Shannon entropy.

## 3.2 LLM Fine-tuning

Recent studies reveal that LLMs possess reasoning apabilities [39], but they often underperform compared to even the simple GNNs when tackling graph learning. The key challenge lies in its inability to directly process the raw graph data and understand topology properties, limiting the generalization ability of LLMs in this domain. To address this, we propose GTA prompts fine-tuning.

This method consists of four distinct fine-tuning instruction types, each designed to enhance structural comprehension, such as local connectivity, node degree, cycle structure, and path-based dependencies, by addressing specific tasks: (1) `Connectivity` involves determining whether or not two nodes in an undirected graph are connected; (2) `Degree` requires the LLM to determine the

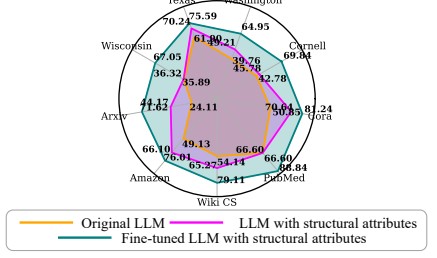

Figure 2: The performance improvements in zero-shot node classification on homophily and heterophily graphs.

degree of a given node based on the adjacency matrix $\mathbf{A}$; (3) `Cycle Detection` requires the LLM to ascertain whether a cycle exists within the given sequence of nodes; (4) `Text Generation` demands

the LLM to generate textual contents of given nodes based on the semantic attributes of preceding nodes in the random walk. Through fine-tuning, LLM exhibits significant improvements on the zero-shot node classification task, as demonstrated in Figure 2. More detailed task descriptions and detailed task-specific GTA prompt templates are provided in Appendix B.3.

### 3.3 GNN-preference-driven Node Selector

After being fine-tuned, the LLM can generate superior node label annotations (as shown in Figure 2). However, how to select nodes for LLM's label annotation to effectively enhance teacher GNNs (those nodes are assumed to be preferred by GNNs) is a challenging problem. Uncertainty is an essential metric for node selection. It mainly consists of two parts: random uncertainty caused by inherent noise and cognitive uncertainty caused by insufficient observation. The former type is inevitable, so we focus on the latter type. From the perspective of collective consensus [40], we design the GNN-preference-driven Node Selector based on the defined $K$-uncertainty ($\delta_K$). Specifically, we measure the cognitive disagreement among the teacher GNNs' SoftMax outputs using the Kullback-Leibler (KL) divergence, and get the preference ranks of all nodes by $\delta_K$, i.e., $\mathcal{V}_{\mathcal{PR}} = \text{Sort}(\{v_1, \ldots, v_N\}, \delta_K(v_1), \delta_K(v_2), \ldots, \delta_K(v_N))$. High $K$-uncertainty of nodes indicates that their prediction uncertainty by GNNs is higher. Those nodes can effectively enhance GNNs if their more accurate labels, annotated by the LLM, are provided to train GNNs, as the following proposition suggests.

**Proposition 3.1.** *These nodes with higher $K$-uncertainty ($\delta_K$) are beneficial for GNNs enhancement.*

$$\delta_K(v) \triangleq \sum_{1 \leq i < j \leq B}^{B} [D_{KL}(f_{T_i}^\theta(v)||f_{T_j}^\theta(v)) + D_{KL}(f_{T_j}^\theta(v)||f_{T_i}^\theta(v))] \propto \delta_v \tag{2}$$

*where $\delta_v$ is the uncertainty of node $v$, is defined as $\frac{1}{B}\sum_{i=1}^{B} D_{KL}(f_{T_i}^\theta(v)||\mathcal{M}(v))$. The $\mathcal{M}(v)$ is the average prediction probability distribution of all $B$ teacher GNNs (See Definition D.1 for details). $D_{KL}(\cdot||\cdot)$ is the function to calculate KL divergence.*

$$f_T^{\theta^*}(\tilde{\mathcal{D}}_L) = \underset{v_i \in \{v_{\mathcal{PR}}^1, v_{\mathcal{PR}}^2, \ldots, v_{\mathcal{PR}}^W | \delta_K(v_{\mathcal{PR}}^W) > \tilde{\delta}_K\}}{\arg\min} \frac{1}{W} \sum \mathcal{L}(f_T^\theta, v_i) \tag{3}$$

*where $\tilde{\mathcal{D}}_L$ is the expanded training dataset. $f_T^{\theta^*}$ is the optimal parameter of teacher GNN. $v_{\mathcal{PR}}^w$ represents the $w$-th nodes in the preference rank. $W$ is the number of selected nodes by GNS and the $\tilde{\delta}_K$ is the $K$-uncertainty threshold depending on the expansion ration.*

The proof is given in Appendix D. By selecting these nodes (illustrated in Figure 3), we ensure that the most uncertain and informative nodes are labeled by the LLM to promote the progress of prediction distillation through the cross-entropy function. Correspondingly, GNS also reduces the inference costs associated with LLMs by not querying all nodes in $\mathcal{D}_U$. To generate high-quality annotations for GNN-preferred nodes, we further design the Distance-based Neighbors Selection (DNS) module, which performs the K-Nearest Neighbor (KNN) search around each selected node across the embedding spaces generated by pretrained teacher GNNs and deletes repeated neighbors. The structure attributes composed of selected neighbors and their textual contents are integrated into the category-induction prompt and inputted into the LLM. Unlike relying solely on neighbors identified by the adjacency matrix (prone to biases from 1-hop homophily), our approach ensures a more robust and diverse selection of high-quality neighbors, facilitating better construction of the category-induction prompt for the LLM. We do not select common KNN neighbors across all the embedding spaces generated by the teacher GNNs, as they may overfit to the adjacency structure.

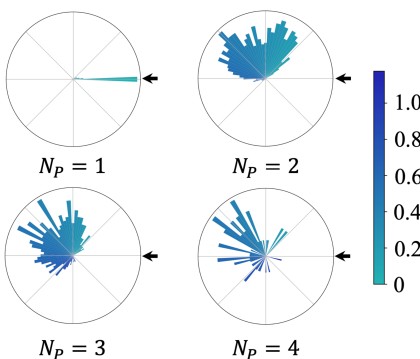

Figure 3: This is exemplified using the CORA dataset. Starting from the arrow and progressing counterclockwise, the KL divergence sum gradually increases, accompanied by a darkening of the triangle colors. The length of each triangle indicates the number of nodes within a specific KL divergence sum range, where $N_P$ denotes the number of classes predicted by the teacher GNNs.

### 3.4 Node-preference-driven GNN Selector

Distilling knowledge simultaneously from multiple teachers to the student is not a good option since nodes with varying local topologies require distinct message-passing modes for optimal representation updates. To achieve this, we introduce the Node-preference-driven GNN Selector (NGS) to select the most appropriate teacher for each node according to the specific attributes and promote tailored knowledge distillation. For each node in the expanded training data (including the initial few labeled nodes and those selected nodes whose labels are annotated by the LLM), we construct a node-specific prompt by combining its semantic, structural, and prediction attributes derived from the enhanced teacher GNNs. This prompt is then inputted to the fine-tuned LLM to determine the most suitable teacher for this node. The GNN selection task is formulated as a reinforcement learning problem that needs to explore the discrete action space and find a series of assignment actions to get the highest global reward across the expanded training data. Through interaction with the training process, the selector progressively refines its decisions on node-to-teacher assignments, leading to a more efficient and effective assignment strategy. Specifically, the fine-tuned LLM, serving as the agent, selects the most appropriate teacher for each node. The policy is trained to maximize classification accuracy on the expanded training data, with the reward tied to the student's performance. To address the non-differentiability of the LLM's decoding process, we add two additional projectors (MLPs) after the logit layer to generate action probabilities and corresponding value estimations, enabling the agent to take discrete teacher-selection actions.

In the RL framework, the elements are structured as (*State, Action, Reward*). During each iteration, the agent interacts with the environment by receiving all attributes of one node in the expanded training data. The agent then takes an action on which teacher is more appropriate.

*State*: Each state corresponds to the prompt $\mathcal{P}_i$ of a node, including node-specific semantic, structural, and prediction attributes. These prompts are detailed in Appendix B.2. The size of the expanded training data is denoted as $W$.

*Action*: The Policy Model (the fine-tuned LLM combined with an MLP projector) generates a text-related output to indicate its selection from multiple teachers, formulated as a probability distribution vector $\boldsymbol{\pi}_T = [\pi_{T_1}, \pi_{T_2}, \ldots, \pi_{T_B}]$, where $\pi_{T_b}$ denotes the probability of selecting the $b$-th teacher $T_b$. The action is determined through sampling.

*Reward*: The function is correlated with the performance of the student GNN, which is trained by distilling knowledge from the selected teacher for each node. The reward function consists of three key parts: classification accuracy, cross-entropy loss, and distillation loss. It can be written as follows:

$$R = \eta * (\mathcal{L}'_{DL} - \mathcal{L}_{CE}) + (1 - \eta) * A_{cc} \tag{4}$$

where $A_{cc}$ represents the classification accuracy of the student GNN on the expanded training data, $\eta$ is a hyper-parameter to balance the three parts, where $\mathcal{L}'_{DL} = -\frac{1}{W} \sum_i^W \widetilde{\mathbf{z}}_i^T \cdot \log f_S^\theta(\mathbf{x}_i)$, and $\mathcal{L}_{CE} = -\frac{1}{Q} \sum_i^Q \mathbf{y}_i \cdot \log f_S^\theta(\mathbf{x}_i)$.

To effectively optimize the agent's actions for better knowledge distillation, we employ the simplified version of Proximal Policy Optimization (PPO) [41] algorithm, which retains the core principles. Specifically, we do not instantiate the Reward Model explicitly and calculate the reward based on the performance of the student GNN. The Reference Model is also not explicitly referenced, because the parameter update objective function we utilize involves a comparison with the previous strategy. To avoid large fluctuations between the current and old policies, we adopt the CLIP strategy [41] to limit the update margin. During the KD process, the parameters $f_A^\theta$ of NGS, remain fixed, while the parameters $f_S^\theta$ of the student GNN are trained. During the NGS process, the parameters $f_S^\theta$ of the student GNN are kept fixed to compute the reward, while the parameters $f_A^\theta$ of NGS based on the collected rewards from all episodes are optimized. The pseudocode, detailed implementations, and time complexity analysis are provided in Appendix C.

## 4 Experiments

### 4.1 Experimental Setup

**Datasets** In order to assess the few-shot node classification performance of our method on TAGs, we conduct a comprehensive series of experiments across 9 real-world datasets: CORNELL, WASH-

Table 1: Node classification accuracies (%) on real-world datasets. $T_1$, $T_2$, $T_3$, and $T_4$ denote the teacher GNNs for homophily or heterophily graphs (refer to the descriptions in **Baselines** for more details of the teacher and student GNNs). The OOM stands for Out-Of-Memory. The best results are highlighted in dark gray, while the runner-up results are marked in light gray.

| Methods | Dataset | CORNELL | WASHINGTON | TEXAS | WISCONSIN | AMAZON RATINGS | OGBN-ARXIV | WIKI CS | PUBMED | CORA |
|---|---|---|---|---|---|---|---|---|---|---|
| | $T_1$ | $58.04_{\pm1.1}$ | $57.84_{\pm2.1}$ | $53.43_{\pm4.1}$ | $59.32_{\pm2.1}$ | $41.22_{\pm6.6}$ | $56.51_{\pm1.2}$ | $81.57_{\pm0.7}$ | $83.34_{\pm2.4}$ | $87.79_{\pm1.6}$ |
| | $T_2$ | $46.29_{\pm0.9}$ | $65.00_{\pm2.5}$ | $82.83_{\pm2.0}$ | $48.30_{\pm2.5}$ | $36.69_{\pm0.2}$ | $59.19_{\pm5.2}$ | $79.08_{\pm1.8}$ | $82.52_{\pm2.1}$ | $87.59_{\pm0.8}$ |
| | $T_3$ | $44.62_{\pm4.3}$ | $55.27_{\pm1.7}$ | $45.19_{\pm2.5}$ | $61.49_{\pm0.6}$ | $37.41_{\pm2.2}$ | $56.71_{\pm3.6}$ | $80.17_{\pm1.6}$ | $79.57_{\pm2.3}$ | $88.38_{\pm0.6}$ |
| | $T_4$ | $32.73_{\pm2.8}$ | $58.33_{\pm1.7}$ | $63.64_{\pm0.1}$ | $62.89_{\pm1.3}$ | $48.93_{\pm0.5}$ | $53.64_{\pm1.3}$ | $72.01_{\pm2.5}$ | $55.15_{\pm1.4}$ | $77.07_{\pm4.0}$ |
| GCNII [29] / # LN 5 | | $57.82_{\pm2.8}$ | $64.17_{\pm3.1}$ | $68.79_{\pm4.3}$ | $60.94_{\pm1.5}$ | $48.22_{\pm4.3}$ | $35.14_{\pm5.6}$ | $58.29_{\pm2.8}$ | $67.83_{\pm7.7}$ | $77.74_{\pm3.7}$ |
| EGNN [30] / # LN 5 | | $53.38_{\pm7.8}$ | $63.33_{\pm1.2}$ | $71.72_{\pm2.9}$ | $55.97_{\pm3.6}$ | $49.03_{\pm8.6}$ | $36.15_{\pm3.9}$ | $63.97_{\pm6.6}$ | $66.12_{\pm9.3}$ | $72.85_{\pm0.7}$ |
| LLMGNN [14] / # LN 5 | | $52.63_{\pm4.3}$ | $41.09_{\pm2.2}$ | $62.82_{\pm3.6}$ | $46.54_{\pm0.9}$ | $47.64_{\pm2.0}$ | $44.11_{\pm2.5}$ | $66.09_{\pm0.4}$ | $78.84_{\pm1.1}$ | $76.23_{\pm1.7}$ |
| GAugLLM [48] / # LN 5 | | $62.98_{\pm3.3}$ | $65.13_{\pm1.1}$ | $73.81_{\pm2.2}$ | $62.20_{\pm0.9}$ | $42.42_{\pm6.0}$ | $53.47_{\pm0.5}$ | $83.10_{\pm1.7}$ | $85.98_{\pm0.6}$ | $79.48_{\pm4.5}$ |
| Self-training [33] / # LN 5 | | $61.90_{\pm6.1}$ | $65.89_{\pm0.5}$ | $72.62_{\pm2.4}$ | $66.29_{\pm0.9}$ | $41.99_{\pm5.0}$ | $33.40_{\pm2.5}$ | $74.99_{\pm0.9}$ | $83.11_{\pm0.4}$ | $83.19_{\pm1.7}$ |
| AGST [31] / # LN 5 | | $71.43_{\pm0.7}$ | $70.09_{\pm0.8}$ | $68.45_{\pm0.8}$ | $70.08_{\pm0.7}$ | $43.11_{\pm0.4}$ | OOM | $72.49_{\pm3.1}$ | $73.75_{\pm0.5}$ | $77.25_{\pm5.6}$ |
| IceBerg [32] / # LN 5 | | $33.33_{\pm11.9}$ | $67.76_{\pm2.9}$ | $50.00_{\pm4.9}$ | $41.53_{\pm2.0}$ | $25.99_{\pm1.5}$ | $33.63_{\pm1.2}$ | $84.88_{\pm0.2}$ | $62.41_{\pm9.3}$ | $76.23_{\pm2.6}$ |
| KDGA [35] | | $54.39_{\pm2.9}$ | $60.00_{\pm0.1}$ | $66.67_{\pm1.5}$ | $58.74_{\pm3.9}$ | $38.06_{\pm1.2}$ | OOM | $65.03_{\pm4.1}$ | OOM | $68.87_{\pm0.8}$ |
| MSKD [36] | | $51.27_{\pm4.2}$ | $50.39_{\pm0.2}$ | $62.63_{\pm2.0}$ | $41.51_{\pm0.2}$ | $35.60_{\pm5.8}$ | $58.27_{\pm1.0}$ | $62.73_{\pm2.9}$ | $45.86_{\pm0.3}$ | $51.61_{\pm0.6}$ |
| BGNN [21] | | $58.60_{\pm3.3}$ | $56.67_{\pm0.8}$ | $65.66_{\pm2.0}$ | $59.12_{\pm6.9}$ | $37.53_{\pm2.0}$ | $46.67_{\pm8.1}$ | $56.96_{\pm4.3}$ | $76.12_{\pm0.7}$ | $71.28_{\pm3.7}$ |
| MTAAM [22] | | $72.68_{\pm1.0}$ | $73.33_{\pm0.8}$ | $80.81_{\pm4.0}$ | $71.69_{\pm1.9}$ | $39.54_{\pm0.2}$ | $32.32_{\pm5.5}$ | $65.24_{\pm3.3}$ | $83.42_{\pm2.3}$ | $79.16_{\pm4.0}$ |
| FairGKD [37] | | $61.05_{\pm2.4}$ | $60.00_{\pm4.1}$ | $84.85_{\pm1.1}$ | $57.11_{\pm0.9}$ | $43.93_{\pm0.5}$ | $42.03_{\pm2.1}$ | $60.25_{\pm1.2}$ | $70.40_{\pm0.3}$ | $69.85_{\pm2.7}$ |
| RANDOM / # LN 5 | | $54.31_{\pm1.7}$ | $58.04_{\pm1.2}$ | $58.93_{\pm1.1}$ | $58.04_{\pm2.7}$ | $57.95_{\pm2.5}$ | $54.97_{\pm3.3}$ | $65.27_{\pm1.8}$ | $66.60_{\pm2.9}$ | $70.64_{\pm2.6}$ |
| VOTING / # LN 5 | | $44.97_{\pm3.0}$ | $58.88_{\pm3.5}$ | $61.31_{\pm2.0}$ | $46.97_{\pm3.6}$ | $58.64_{\pm2.1}$ | $58.53_{\pm2.0}$ | $72.28_{\pm2.2}$ | $70.64_{\pm3.1}$ | $74.32_{\pm3.1}$ |
| PKD$_{\text{Llama}}$ | # LN 1 | $74.60_{\pm2.1}$ | $76.64_{\pm0.9}$ | $80.36_{\pm1.3}$ | $69.32_{\pm2.8}$ | $64.11_{\pm1.7}$ | $53.67_{\pm1.6}$ | $79.31_{\pm0.8}$ | $83.75_{\pm1.1}$ | $85.64_{\pm2.1}$ |
| | # LN 3 | $76.72_{\pm0.9}$ | $81.36_{\pm1.0}$ | $83.33_{\pm0.7}$ | $71.49_{\pm1.5}$ | $65.64_{\pm0.9}$ | $58.65_{\pm2.2}$ | $80.01_{\pm0.6}$ | $84.34_{\pm0.9}$ | $86.18_{\pm1.7}$ |
| | # LN 5 | $80.95_{\pm1.1}$ | $83.74_{\pm0.4}$ | $86.31_{\pm0.5}$ | $76.89_{\pm0.9}$ | $66.79_{\pm0.3}$ | $61.03_{\pm0.7}$ | $81.39_{\pm0.4}$ | $85.69_{\pm0.3}$ | $91.14_{\pm0.3}$ |

INGTON, TEXAS, WISCONSIN [25], AMAZON RATINGS [42], OGBN-ARXIV [43], WIKI CS [44], PUBMED, CORA [45]. They have various 1-hop homophily ratios [46] and additional details of the datasets can be found in Appendix A. For the KD-baselines, we partition the nodes of each graph into training, validation, and test sets, allocating 48%, 32%, and 20%, respectively, based on the proportion division mentioned in [47]. For PKD and other baselines, we randomly select 1, 3, and 5 labeled nodes per class as the initial training data and then expand the dataset to 48% of the total using the GNS module. The remaining data is randomly split into 32% for validation and 20% for testing, with the preserved indices for the baselines. This operation is repeated 5 times. We report the average test classification accuracy and standard deviation of each model with parameters that lead to the peak validation accuracy.

**Baselines**   We compare our method against the following baseline models: (i) Advanced GNNs: GCNII [29] and EGNN [30]; (ii) GNNs enhanced by LLMs: LLMGNN [14] and GAugLLM [48]; (iii) self-training for graph learning: Self-training [33], AGST [31] and IceBerg [32]; (iv) Knowledge Distillation (KD) for GNNs: KDGA [35], MSKD [36], BGNN [21], MTAAM [22], and FairGKD [37]. For homophily graphs, the teacher GNNs used are: GCN [4] ($T_1$), GAT [23] ($T_2$), APPNP [24] ($T_3$), H$_2$GCN [25] ($T_4$), and the student is GCN; for heterophily graphs, the teacher GNNs employed are: DirGNN [28] ($T_1$), GPRGNN [26] ($T_2$), HoloNets [27] ($T_3$), H$_2$GCN ($T_4$) and the student is H$_2$GCN. The LLM used in the experiments is Llama-3.1-8B-Instruct [49].

## 4.2   Performance Analysis and Discussion

Notably, **# LN 1, # LN 3, # LN 5** indicate only 1, 3, 5 labeled nodes per class are used for training PKD, while the results of the teacher GNNs ($\{T_i\}_{i=1}^4$) and other baselines are trained under the data splitting of 48%/32%/20% as mentioned above. According to Table 1, our method almost achieves the best or second-best accuracy results.

Due to the extreme insufficiency of labels, GCNII and EGNN are restricted in further improvement, although they have distinctive network architectures. Lacking carefully designed fine-tuning and enough cognition makes LLMGNN fail to produce high-quality pseudo labels and is dramatically defeated by our method PKD. Although GAugLLM harnesses LLM for feature and structure augmentations to benefit GNN, its self-training depends only on SoftMax scores to identify candidate nodes to assign pseudo-labels, a method that can sometimes be unreliable. GAugLLM achieves the best result on the PUBMED dataset, but it is outperformed by PKD on other datasets. AGST is excessively dependent on the original graph topology for label propagation, rendering it vulnerable

to structural noise and facing significant challenges when transferred to large-scale graphs, such as OGBN-ARXIV. IceBerg does not perform well on heterophily graphs because its capacity to disseminate information across longer distances is hampered by the proliferation of noise edges. MTAAM shows satisfactory performance on most datasets, due to its ability to autonomously identify the most valuable knowledge from each teacher during training. FairGKD achieves runner-up results on some datasets. The poor performances of KDGA and BGNN result from their excessive sensitivity to GNN selection. MSKD is equipped with the fixed message-passing mechanism, showing that the single message-passing mechanism underperforms on all the datasets compared to PKD. The RANDOM / **# LN 5** approach refers to randomly selecting node predictions from 4 teachers, utilizing 5 labeled nodes per class to train teacher GNNs. The VOTING / **# LN 5** method selects the most frequently predicted label from 4 teachers as the annotation label. We can see that these two simple and intuitive strategies are defeated by PKD on all datasets.

Our PKD consistently achieves superior node classification results across all datasets, irrespective of the specific type of LLM. The few-shot node classification results after replacing Llama-3.1-8B-Instruct with Qwen2.5-7B-Instruct [50] and Mixtral-7B-Instruct-v0.3 [51] are shown in Table 2.

Table 2: Few-shot node classification accuracy (%) on eight TAGs using three different LLMs. The **# LN 1, # LN 3, # LN 5** represent 1, 3, 5 labeled nodes per class, respectively. The best results are highlighted in dark gray, while the runner-up results are marked in light gray.

| Methods | Dataset | CORNELL | WASHINGTON | TEXAS | WISCONSIN | AMAZON RATINGS | OGBN-ARXIV | WIKI CS | PUBMED | CORA |
|---|---|---|---|---|---|---|---|---|---|---|
| PKD$_{Qwen}$ | # LN 1 | $73.54_{\pm2.6}$ | $75.70_{\pm1.1}$ | $82.14_{\pm0.8}$ | $72.59_{\pm1.3}$ | $74.58_{\pm1.1}$ | $54.17_{\pm2.2}$ | $79.49_{\pm1.2}$ | $82.81_{\pm0.8}$ | $86.45_{\pm1.0}$ |
| | # LN 3 | $77.25_{\pm1.4}$ | $77.35_{\pm0.9}$ | $84.52_{\pm0.4}$ | $73.86_{\pm0.7}$ | $75.46_{\pm0.8}$ | $60.63_{\pm1.0}$ | $80.01_{\pm0.6}$ | $83.61_{\pm1.1}$ | $87.74_{\pm0.8}$ |
| | # LN 5 | $79.84_{\pm0.6}$ | $79.63_{\pm0.6}$ | $85.71_{\pm0.2}$ | $74.24_{\pm0.2}$ | $77.69_{\pm0.6}$ | $62.62_{\pm2.1}$ | $81.21_{\pm0.2}$ | $85.96_{\pm0.6}$ | $90.07_{\pm0.4}$ |
| PKD$_{Mixtral}$ | # LN 1 | $76.31_{\pm2.2}$ | $74.42_{\pm1.5}$ | $79.41_{\pm3.3}$ | $69.81_{\pm0.5}$ | $70.02_{\pm1.2}$ | $57.69_{\pm0.4}$ | $80.56_{\pm0.8}$ | $82.42_{\pm0.9}$ | $84.87_{\pm2.4}$ |
| | # LN 3 | $78.95_{\pm1.2}$ | $76.74_{\pm3.1}$ | $82.86_{\pm1.7}$ | $75.47_{\pm0.8}$ | $71.50_{\pm2.6}$ | $61.17_{\pm0.6}$ | $81.96_{\pm1.3}$ | $83.19_{\pm2.7}$ | $87.64_{\pm1.1}$ |
| | # LN 5 | $81.58_{\pm2.1}$ | $81.39_{\pm2.5}$ | $85.29_{\pm1.9}$ | $77.36_{\pm3.1}$ | $73.96_{\pm1.9}$ | $62.44_{\pm0.6}$ | $83.33_{\pm1.4}$ | $84.71_{\pm1.6}$ | $88.56_{\pm0.7}$ |

Furthermore, to evaluate the quality of LLM-generated pseudo-labels, we compare the node classification performance of PKD and three baselines under different label settings (# LN 5, 48% training ratio expanded by the annotated labels and real labels, respectively). The experiments are conducted on four datasets (CORA, WIKI CS, WASHINGTON, and WISCONSIN). The results are presented in Table 3. For GCNII and IceBerg, they are proposed to tackle the challenge of sparse labels, using the LLM-annotated node labels can improve their performance on all datasets. However, using the same number of real labels achieves better performance.

Table 3: Classification accuracy comparison under different label configurations. The best results are highlighted in dark gray, while the runner-up results are marked in light gray.

| Models | Labels configuration | CORA | WIKI CS | WASHINGTON | WISCONSIN |
|---|---|---|---|---|---|
| GCNII | # LN 5 | 77.74 | 56.29 | 64.17 | 60.94 |
| | 48% LLM-generated labels | 76.69 | 51.18 | 70.83 | 62.50 |
| | 48% real labels | 81.54 | 59.17 | 71.79 | 65.98 |
| IceBerg | # LN 5 | 76.23 | 84.88 | 67.76 | 41.53 |
| | 48% LLM-generated labels | 78.66 | 71.23 | 70.12 | 42.22 |
| | 48% real labels | 81.94 | 86.49 | 72.04 | 45.43 |
| MSKD | # LN 5 | 43.91 | 46.81 | 45.29 | 33.33 |
| | 48% LLM-generated labels | 45.89 | 54.06 | 48.17 | 39.50 |
| | 48% real labels | 51.61 | 62.73 | 50.39 | 41.51 |
| PKD$_{Llama}$ | # LN 5 | 90.27 | 81.39 | 83.74 | 76.89 |

## 4.3 Ablation Study

Generally, the fine-tuned LLM using our proposed GTA prompts also demonstrates pretty zero-shot node classification performance, surpassing some semi-supervised GNNs from the values in Figure 2.

We assess the significance of the GTA prompts, DNS and $\mathcal{V}_{\mathcal{PR}}$ with the following default parameter settings: **# LN** = 3, $K$ = 4. Here, $K$ denotes the number of selected neighbors surrounding the node, to be annotated, within each embedding space of the teacher GNNs structure attributes. In the absence of DNS, neighbors are selected according to the adjacency matrix directly; in the non-use of $\mathcal{V}_{\mathcal{PR}}$, we expand the training data by random selection.

Table 4: Ablation study for GTA, DNS, and $\mathcal{V}_{\mathcal{PR}}$. ⇑ denotes an accuracy (%) increment. The three components play different roles in the improvement of the performance of our method.

| Dataset/Modeule | GTA | DNS | $\mathcal{V}_{\mathcal{PR}}$ | Accuracy | Dataset/Module | GTA | DNS | $\mathcal{V}_{\mathcal{PR}}$ | Accuracy |
|---|---|---|---|---|---|---|---|---|---|
| **CORA** | ✗ | ✗ | ✗ | 45.02 | **AMAZON RATINGS** | ✗ | ✗ | ✗ | 42.01 |
| | ✗ | ✗ | ✓ | ⇑ 26.94 | | ✗ | ✗ | ✓ | ⇑ 13.02 |
| | ✗ | ✓ | ✓ | ⇑ 30.99 | | ✗ | ✓ | ✓ | ⇑ 16.96 |
| | ✓ | ✓ | ✓ | ⇑ 41.14 | | ✓ | ✓ | ✓ | ⇑ 23.97 |

As shown in Table 4, the implementations of GAT prompts, DNS, and $\mathcal{V}_{\mathcal{PR}}$ result in varying degrees of performance improvement. Supported by fine-tuning with GTA prompts, the LLM's enhanced logical reasoning ability, combined with high-quality neighboring nodes, substantially enhances zero-shot node classification capability, leading to superior classification performance improvement. Additionally, we also assess the methods without using reinforcement learning in the teacher selection process, including entropy-based ranking, i.e., selecting the teacher GNN with the highest prediction confidence, random selection, and end-to-end learning. Their relevant results are provided in the Appendix E.2.

To assess the effectiveness of each part in the reward function (Eqn. (4)), we visualize the training processes of three variants in Figure 4: (a) $R_1$: The reward function for teacher GNN selection depends solely on the classification accuracy of the student GNN on the expanded training data; (b) $R_2$: In addition to classification accuracy, the reward function also incorporates the negative cross-entropy loss ($-\mathcal{L}_{CE}$); (c) $R_3$: Building upon $R_2$, the reward function also includes the negative knowledge distillation loss ($-\mathcal{L}_{DL}$). As shown in Figure 4, both the three parts contribute to the improved classification performance.

Figure 4: The comparison of different Rewards. When including all three parts simultaneously, our method (the curve in green) performs the best.

## 4.4 Sensitivity Analysis

We investigate the impact of the hyperparameter $K$ on the zero-shot node classification performance. We vary the value of $K$ within the range $\{1, 2, 3, 4, 5\}$ for homophily graphs and heterophily graphs to observe the variation in zero-shot node classification accuracy. As illustrated in Figure 5, accuracy exhibits significant fluctuations as $K$ changes. When $K = 4$, the fine-tuned LLM demonstrates strong performance on most graphs.

To further explore the relationship between the parameter scale of LLM and PKD's performance, we evaluated Qwen2.5-7B-Instruct with three different parameter scales: 7B/14B/32B parameters. The results are shown in Table 5. Obviously, the classification performance of PKD basically gets better with the increase of parameter scale. This is mainly related to the LLMs with larger parameter-scale have richer knowledge storage and better ability handling complex tasks.

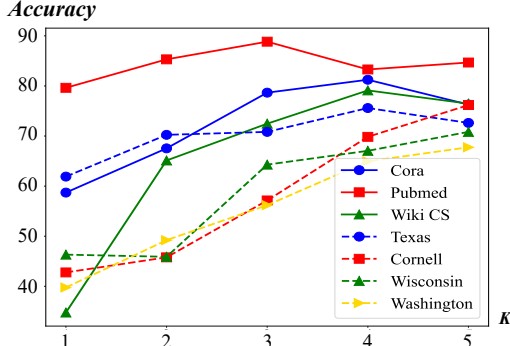

Figure 5: The effects of $K$ on homophily and heterophily graphs. When $K = 4$, zero-shot node classification accuracy of the fine-tuned LLM is the highest on most graphs.

Next, we investigate the ratios of nodes selected for annotating their labels by the LLM as a means to expand the training set. The results are given in Table 6. Increasing the expansion ratio can enhance the performance of PKD. This improvement can be attributed not only to the high-quality label annotation generated by the fine-tuned LLM, but also to the characteristic of PKD that is underpinned by selecting each node the most appropriate teacher GNN for knowledge distillation.

Furthermore, we perform the hyperparameter sensitivity analysis over the loss-weight coefficients $\alpha, \beta, \gamma, \eta$. For the sensitivity analysis of $\alpha$, we set $\beta = 1, \gamma = 1, \eta = 0.5$.

Table 5: The few-shot node classification accuracy (%) of PKD with different parameter-scale LLM.

| Datasets | CORA | | |
|---|---|---|---|
| Parameter scales | 7B | 14B | 32B |
| PKD$_{\text{Qwen}}$ | 90.07 | 90.58 | 91.54 |
| Datasets | PUBMED | | |
| Parameter scales | 7B | 14B | 32B |
| PKD$_{\text{Qwen}}$ | 85.96 | 86.64 | 87.16 |

This strategy also applies to the sensitivity analysis of $\beta$ and $\gamma$. As for $\eta$, we fix $\alpha, \beta$, and $\gamma$ to their respective best values. The $\beta$ and $\eta$ are more sensitive than the other two hyperparameters, because $\beta$ is related to the supervision signal provided by the LLM-generated annotation labels and $\eta$ is related to the student GNN's performance. All the results are reported in the Appendix E.3.

Table 6: The results with different node annotation ratios. The best results are highlighted in dark gray, while the runner-up results are marked in light gray.

| Node annotation ratios | 10% / # LN 5 | 20% / # LN 5 | 30% / # LN 5 | 40% / # LN 5 | 48% / # LN 5 |
|---|---|---|---|---|---|
| AMAZON RATINGS | $50.27_{\pm 6.6}$ | $55.91_{\pm 1.8}$ | $62.07_{\pm 3.5}$ | $63.93_{\pm 1.1}$ | $66.79_{\pm 0.3}$ |
| CORA | $73.37_{\pm 5.1}$ | $77.51_{\pm 1.1}$ | $81.49_{\pm 2.5}$ | $83.27_{\pm 2.2}$ | $91.14_{\pm 0.3}$ |

## 4.5 Running Time

We also study the training efficiencies of PKD and all baselines. The running times on CORA are shown in Table 7. There is a trade-off between accuracy and time complexity. The incorporation of the LLM undoubtedly boosts the few-shot classification accuracy of GNNs on TAGs, but the training time increases. When applied to the bigger graphs, the time increase will be more obvious.

Table 7: Running time (second per epoch) of each method, including the pretraining process.

| Datasets / Methods | $T_1$ | $T_2$ | $T_3$ | $T_4$ | GCNII | EGNN | LLMGNN | GAugLLM | – |
|---|---|---|---|---|---|---|---|---|---|
| CORA | 0.006 | 0.197 | 0.247 | 0.035 | 0.014 | 0.366 | 0.630 | 0.402 | – |

| Datasets / Methods | Self-training | AGST | IceBerg | KDGA | MSKD | BGNN | MTAAM | FairGKD | PKD$_{\text{Llama}}$ |
|---|---|---|---|---|---|---|---|---|---|
| CORA | 0.016 | 0.018 | 0.011 | 3.911 | 2.289 | 1.001 | 3.318 | 4.100 | 7.314 |

## 5 Conclusions, Limitations & Future Work

In this work, we have proposed a preference-driven knowledge distillation (PKD) framework for few-shot node classification on TAGs, consisting of GNN-preference-driven Node Selector (GNS) and Node-preference-driven GNN Selector (NGS). Fine-tuned with our proposed GTA prompts, the refined LLM generates high-quality annotations. The GNS effectively determines nodes for the fine-tuned LLM to annotate and promotes knowledge distillation from the LLM to teacher GNNs. The NGS tailors for each node the most appropriate message-passing mechanism, promoting knowledge distillation from teacher GNNs to the student GNN. On various real-world TAGs, our method PKD outperforms almost all advanced GNNs and KD methods for few-shot node classification while using only a few node labels. One limitation of our method is that it is designed for TAGs. Moving forward, we plan to further explore more efficient mechanism of synergizing LLM and GNN to address the limitation of training efficiency as well as datasets beyond TAGs.

## 6 Acknowledgments and Disclosure of Funding

We thank the anonymous reviewers for their valuable and constructive comments. This work was supported partially by the National Natural Science Foundation of China under Grants 62176184, 62476109, and 62206108, and the Fundamental Research Funds for the Central Universities.

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

# A  Detailed Description of Datasets

Table 8: Statistics of datasets. The **Hom. ratio** means 1-hop homophily ratio.

| Dataset | CORNELL | WASHINGTON | TEXAS | WISCONSIN | AMAZON RATINGS | OGBN-ARXIV | WIKI CS | PUBMED | CORA |
|---|---|---|---|---|---|---|---|---|---|
| **Hom. ratio** | 0.1504 | 0.1545 | 0.1989 | 0.2109 | 0.4777 | 0.6542 | 0.6588 | 0.7924 | 0.8252 |
| **# Node** | 189 | 214 | 168 | 264 | 5068 | 169343 | 11701 | 19717 | 2708 |
| **# Edge** | 166 | 182 | 91 | 388 | 17334 | 1166243 | 216123 | 88648 | 10556 |
| **# Features** | 1703 | 1703 | 1703 | 1703 | 300 | 128 | 300 | 500 | 1433 |
| **# Classes** | 5 | 5 | 5 | 5 | 5 | 40 | 10 | 3 | 7 |
| **Domain** | Web page | Web page | Web page | Web page | Co-purchase | Co-citation | Wikipedia page | Co-citation | Co-citation |

**CORNELL, WASHINGTON, TEXAS, and WISCONSIN:**

These four datasets are derived from the WEBKB webpage dataset, collected from the computer science departments of various universities. In these datasets, nodes represent web pages, while edges denote hyperlinks connecting them. All words from the given web pages are collected as the features for the nodes. The webpage categories can be listed as following: Student, Project, Course, Staff, Faculty.

**AMAZON RATINGS:**

This dataset is derived from the AMAZON product co-purchasing network metadata, sourced from the SNAP datasets [52]. Nodes represent products (Books, Music CDs, DVDs, Videos) and edges signify relationships between products that are frequently co-purchased. The task involves predicting the average rating assigned to each product by reviewers. The possible rating values are grouped into five distinct classes. For node features, we utilize the NV-Embed-v2 [53] embeddings generated from the product descriptions. To reduce the size of the graph, we only consider the largest connected component of the 5-core of the graph.

**WIKI CS:**

WIKI CS is a graph derived from the Wikipedia platform. The nodes in WIKI CS represent Wikipedia page descriptions, while the edges correspond to hyperlinks between distinct pages. The WIKI CS dataset and its raw text [44] are sourced from OFA [54]. The graph consists of 11,701 nodes and 216,123 edges. The WIKI CS dataset is suitable for node classification tasks. The WIKI CS dataset is categorized into 10 distinct categories: Computational Linguistics, Databases, Operating Systems, Computer Architecture, Computer Security, Internet Protocols, Computer File Systems, Distributed Computing Architecture, Web Technology, Programming Language Topics.

**CORA, PUBMED, and OGBN-ARXIV:**

The CORA dataset represents a co-citation graph of computer science research papers. The dataset is sourced from OFA [54], with the original data derived from [9]. In [9], the authors recollect the dataset due to the commonly employed bag-of-words features in the widely used CORA dataset within the GNN community, where raw text is difficult to retrieve. The revised CORA dataset contains 2,708 nodes and 10,556 edges, matching the specifications of the original dataset. The dataset is divided into 7 categories: Theory, Reinforcement Learning, Genetic Algorithms, Neural Networks, Probabilistic Methods, Case-Based, Rule Learning.

The PUBMED dataset represents a co-citation graph of biomedical research papers focused on diabetes mellitus. The source and processing procedure of PUBMED are identical to those of CORA. After processing, the dataset consists of 19,717 nodes and 88,648 edges. The dataset is classified into 3 categories: Experimentally Induced Diabetes, Type 1 Diabetes, Type 2 Diabetes.

The OGBN-ARXIV dataset is a citation graph of papers from the arXiv platform. It is collected from the Arxiv dataset and its raw text as OGB[43] and OFA [54]. There are 169,343 nodes and 1,166,243 edges in the graph. It contains 40 sub-categories of compute science.

# B Detailed Prompts

We provide all specific prompt templates in the following for zero-shot node classification, Node-preference-driven GNN Selector and GTA Prompts, respectively.

## B.1 Prompts for Zero-shot Node Classification

The complete prompts for zero-shot node classification are provided as below. Similarly, for each dataset, we refine specific descriptions to ensure contextual coherence.

Table 9: The prompt template for zero-shot node classification.

| Role | Prompt |
|------|--------|
| System Prompt | Papers in this field can be divided into 7 categories: [Case Based, Genetic Algorithms, Neural Networks, Probabilistic Methods, Reinforcement Learning, Rule Learning, Theory]. You will serve as an assistant to help me to classify this target paper into the 7 categories above according to its description and related papers' descriptions, who may be of the same category as this target paper. I will provide you with the descriptions of this target paper and its related papers. 

 Here are the instructions: 
 I will provide you with information in the form of a JSON string that describes the target paper: 
 ***Title:*** the title of this target paper. ***Abstract:*** the abstract of this target paper. 
 ***Related Title:*** the title of the related paper. ***Related Abstract:*** the abstract of the related paper. 
 ***Related Title:*** the title of the related paper. ***Related Abstract:*** the abstract of the related paper. 
 ...... 

 Requirements: 
 ❶ Please provide your response in JSON format, following this structure: 
 **Reasoning:** Briefly explain your reasoning process for the predicted category. 
 **Category:** The best category you predict for this paper, this category must belong to these 7 categories: [Case Based, Genetic Algorithms, Neural Networks, Probabilistic Methods, Reinforcement Learning, Rule Learning, Theory]; 
 ❷ There are 2000 words limits for the reasoning; 
 ❸ Do not provide any other text outside the JSON string; 
 ❹ Focus only on content in the actual text and avoid making false associations; 
 ❺ The output can only contain category and reasoning. |
| User Prompt | ***Title:*** $t_{title}$. ***Abstract:*** $t_{abstract}$. 
 ***Related Title:*** $t_{title}^{r_1}$. ***Related Abstract:*** $t_{abstract}^{r_1}$. 
 ***Related Title:*** $t_{title}^{r_2}$. ***Related Abstract:*** $t_{abstract}^{r_2}$. 
 ***Related Title:*** $t_{title}^{r_3}$. ***Related Abstract:*** $t_{abstract}^{r_3}$. 
 ... |

## B.2 Prompts for Node-preference-driven GNN Selector

Unlike the prompts used zero-shot node classification described above, we do not collect responses from the LLM; instead, we focus solely on the outputs generated by the subsequent projector. Similarly, for each dataset, we refine certain descriptions to maintain contextual consistency.

Table 10: The prompt template for Node-preference-driven GNN Selector.

| Role | Prompt |
|------|--------|
| System Prompt | There are four names of teacher networks: [APPNP, GCN, $H_2$GCN, GAT]. We need to perform knowledge distillation for each node in this graph consist of nodes (papers) and edges (citation relationships). You will serve as an assistant to help me to assign the best teacher network for the target node (paper) based on the following information. I will provide you with three kinds of attributes of the target node (paper). 

 Here are the instructions: 
 I will provide you with information in the form of a `JSON` string that describes the node (paper): 
 *Semantic attributes:* the title and abstract of this paper. 
 *Structure attributes:* four teacher networks' logit output of this target node. 
 *Prediction attributes:* important neighbors (papers), which are closely related the target node (paper) and their contents. 

 Requirements: 
 ❶ Please provide your response in `JSON` format, following this structure: 
 **Reasoning:** Briefly explain your reasoning process for the selected teacher network. 
 **Teacher network:** The best teacher network you assign for this node (paper), this result must belong to these 4 teachers: [APPNP, GCN, $H_2$GCN, GAT]; 
 ❷ There are 2000 words limits for the reasoning; 
 ❸ Do not provide any other text outside the `JSON` string; 
 ❹ Focus only on content in the actual text and avoid making false associations; 
 ❺ The output can only contain teacher network and reasoning. |
| User Prompt | *Semantic attributes:* It is the content description of this target paper: $\mathbf{t}$. 
 *Structure attributes:* It has following important neighbors (papers), which are closely related the target paper. Their content descriptions are: ... 
 *Prediction attributes:* 
 The APPNP's logits output of this target paper is str($\mathbf{z}_{APPNP}$), 
 The GCN's logits output of this target paper is str($\mathbf{z}_{GCN}$), 
 The $H_2$GCN's logits output of this target paper is str($\mathbf{z}_{H_2GCN}$), 
 The GAT's logits output of this target paper is str($\mathbf{z}_{GAT}$) 
 ... |

## B.3 Graph Topology Aware (GTA) Prompts

Generating effective prompts for graph-based tasks can be challenging for LLMs, due to the inherent complexity of graph structures and relationships that must be accurately represented. To address this challenge, we propose structured-tasks text for graph topology aware, designed specifically for fine-tuning LLMs.

**TASK 1: `Connectivity`** This task is determining whether or not two nodes $v_i$ and $v_j$ in an undirected graph are connected. Specifically, we randomly select node pairs $v_i, v_j \in \mathcal{V}$ and ask whether or not an edge exists between them in the graph, answering with a "True/False" response. To ensure prompt diversity, only one-third of the possible node pairs are selected for each graph.

**TASK 2: `Degree`** The degree of a node, $D$, is the number of nodes directly connected to it. In this task, we group nodes based on their degree and select a node $v_i$ from a group. The LLM is then given the node's local structure according to the adjacency matrix $\mathbf{A}$, and is asked for the degree of the node. To prevent repetitive prompts, only one-third of the nodes from each degree group are selected.

**TASK 3: `Cycle Detection`** A cycle in an undirected graph without self-loop is a path where the first and last nodes are the same. This task requires the LLM to answer whether a cycle exists in the given sequence of nodes, $\{v_1, ..., v_l, ..., v_1\}$. We generate random walks [55] of length greater than

10 and arrange them into node sequences. After describing their neighbors information (derived from the adjacency matrix $\mathbf{A}$), the LLM is then asked whether or not any sequence of nodes forms a cycle.

**TASK 4: Text Generation** We randomly select a node set $\mathcal{W} = \{v_i\}_{i=1}^{N/3}$ as the source nodes, and a breadth-first search (BFS) is conducted from each source node to identify nodes in graph at a distance greater than $t$ edges from $v_i$, which are collected as target nodes. Redundant nodes are removed via the long-to-short path conversion module [56]. The LLM is tasked with generating textual descriptions of target nodes based on the semantic attributes of the preceding nodes in the path.

Specifically, **TASK 1** enhances the LLM's ability to identify neighboring nodes and understand the structure of local neighborhoods; **TASK 2** strengthens the LLM's ability to recognize the significance of node degrees within the graph context; **TASK 3** reinforces the LLM to reason about complex graph topologies, such as cycles and long-range node dependencies; **TASK 4** improves path-based reasoning and contextualization of nodes in the local graph structure.

Table 11: The prompt template for the TASK 1: `Connectivity`.

| Role | Prompt |
|---|---|
| System Prompt | You will serve as a graph machine learning expert in connectivity detection to help me to determine whether the edge exists between the given two targeted nodes. There is a undirected graph consisting of papers (nodes) and the citation relationships (edges) between them. I will provide the information of the two targeted nodes and their neighbors, consisting of indexes, textual content. 

 Here are the instructions: 
 I will provide you with information in the form of a `JSON` string that describes the target papers: 
 ***The first targeted paper:*** 
 Node index: ...; Title: ...; Abstract: ...; 
 The $k_{th}$ neighbor: Index:...; Title: ...; Abstract: ...; 

 ... 

 ***The second targeted paper:*** 
 Node index: ...; Title: ...; Abstract: ...; 
 The $k_{th}$ neighbor: Index:...; Title: ...; Abstract: ...; 

 ... 

 Requirements: 
 ❶ Please provide your response in `JSON` format, following this structure: 
 **Reasoning:** Briefly explain your reasoning process for the selected teacher network. 
 **Answer:** You only can select one from [True, False] as the best answer; 
 ❷ There are 2000 words limits for the reasoning; 
 ❸ Do not provide any other text outside the `JSON` string; 
 ❹ Focus only on content in the actual text and avoid making false associations; 
 ❺ The output can only contain answer and reasoning. |
| User Prompt | ***The first targeted paper:*** 
 Node index: $i$; Title: $\mathbf{t}^i_{title}$; Abstract: $\mathbf{t}^i_{abstract}$ 
 The $k_{th}$ neighbor's node index: $I^i_k$ Title: $\mathbf{t}^{I^i_k}_{title}$ Abstract: $\mathbf{t}^{I^i_k}_{abstract}$... 
 ... 
 ***The second targeted paper:*** 
 Node index: $j$; Title: $\mathbf{t}^j_{title}$; Abstract: $\mathbf{t}^j_{abstract}$ 
 The $k_{th}$ neighbor's node index: $I^j_k$ Title: $\mathbf{t}^{I^j_k}_{title}$ Abstract: $\mathbf{t}^{I^j_k}_{abstract}$ ... 
 ... |

The full prompts for `Connectivity` is presented above. When generating prompts for different datasets, we adjust certain descriptions to better align with the specific context. For example, when constructing prompts for TEXAS, the background description should be adapted to reflect web pages, and the relationship should be revised to hyperlinks, along with other context-specific adjustments.

Similarly, for each task, the prompts must also be modified to correspond to the specific content described in Section 3.2.

## C  Implementation Details and Time Complexity Analysis

---

**Algorithm 1:** The training of PKD.

---

**Input:** $\mathcal{G}_T = (\mathcal{V}, \mathcal{E}, \mathbf{X}, \mathbf{A}, \mathbf{T})$, training dataset with true labels $\mathcal{D}_L$, teacher GNNs $\{T_b\}_{b=1}^4$ with parameters $\{f_{T_b}^\theta\}_{b=1}^4$, student GNN $S$ with parameter $f_S^\theta$, fine-tuned LLM $LLM^\theta$, Policy Model $f_A^\theta$, Value Model $f_V^\phi$, epoch number of RL $L_1$

**Output:** The expanded training dataset $\tilde{\mathcal{D}}_L$, optimized parameters $LLM^{\theta^*}, f_S^{\theta^*}, f_A^{\theta^*}, f_V^{\phi^*}$ and predicted labels $\tilde{y}$.

1 $\tilde{\mathcal{D}}_L \leftarrow \mathcal{D}_L$;
2 $LLM^{\theta^*} \leftarrow LLM^\theta$;
3 Filter out GNN-preference nodes based on the preference rank $\mathcal{V}_{\mathcal{PR}}$ and get their annotations from $LLM^{\theta^*}$;
4 Conduct prediction distillation from $LLM^{\theta^*}$ to $\{f_{T_b}^\theta\}_{b=1}^4$ for retrain them ;
5 **for** $l_1 \leftarrow 1$ **to** $L_1$ **do**
6      Shuffle $\tilde{\mathcal{D}}_L$ to get a new training sequence;
7      Complete prompts $\{\mathcal{P}_i\}_{i=1}^W$ for each selected nodes;
8      **for** *each node $v_{\mathcal{PR}} \in \tilde{\mathcal{D}}_L$* **do**
9          NSG select teacher GNN for $v_{\mathcal{PR}}$ and get one-hot vector $\mathbf{m}_i$;
10          Update the parameter $f_S^\theta$ and get reward $R_i$ by Eqn. (1);
11          Store $(\mathcal{P}_i, \mathbf{m}_i, R_i)$ to the episode history $\mathcal{F}$;
12      Update the parameter $f_A^\theta$ and $f_V^\phi$ by Eqn. (5) and Eqn. (6) ;
13 **return** $\tilde{\mathcal{D}}_L, LLM^{\theta^*}, f_S^{\theta^*}, f_A^{\theta^*}, f_V^{\phi^*}$;

---

First of all, we outline the training setup employed for the experiments detailed in Section 4.2. Uniform training hyper-parameters are applied across all baseline models and datasets. Specifically, the following hyper-parameter values are utilized: the hidden dimension is set to 128. We use ReLU activation functions in all our baseline models. The Adam optimizer is utilized with a learning rate of $1 \times e^{-2}$ and weight decay of $5 \times e^{-4}$. We train each baseline for 600 steps and select the best step based on the validation accuracy. In our proposed method, we train the student 5 epochs after GNN selection driven by node attributes every time and train the agent 200 epochs. The other weight hyper-parameters are set as follows: $\alpha = 0.5, \beta = 1, \gamma = 0.1, \eta = 0.3, c_1 = 0.5, c_2 = 0.01, \epsilon = 0.2$.

Additionally, the parameters of Action Model and Value Model are updated as follows:

$$f_A^\theta \leftarrow f_A^\theta - \rho_A \nabla_{f_A^\theta}(\mathcal{L}_A + c_1 \mathcal{L}_V - c_2 H(\boldsymbol{\pi}_T)) \tag{5}$$

$$f_V^\phi \leftarrow f_V^\phi - \rho_V \nabla_{f_V^\phi} \mathcal{L}_V \tag{6}$$

where $f_A^\theta$ and $f_V^\phi$ represent the trainable parameters of the Policy Model and Value Model, respectively. $\rho_A$ and $\rho_V$ are their learning rates and $\nabla_{f_A^\theta}$ and $\nabla_{f_V^\phi}$ are the gradients of their parameters. $\mathcal{L}_A$ and $\mathcal{L}_V$ are objective functions belonging to the Policy Model and Value Model, respectively. $c_1, c_2$ are hyper-parameters to balance weights. $H(\boldsymbol{\pi}_T)$ is employed to enhance the entropy of the policy and promote sufficient exploration. Based on the CLIP strategy [41], the final objective function of the Policy Model is:

$$\mathcal{L}_A = -\mathbb{E}_i[\min(r_i(f_A^\theta)\hat{A}_i, \text{clip}(r_i(f_A^\theta), 1 - \epsilon, 1 + \epsilon)\hat{A}_i)] \tag{7}$$

where $\mathbb{E}_i$ represents the expectation in the time step $i$. $r_i(f_A^\theta)$ is the ratio of the $i$-th policy to the $(i-1)$-th policy. $\hat{A}_i$ is the advantage estimation in the current step, denoting how good or bad the *Action* is. $\epsilon$ is a hyper-parameter, which determines the range of the CLIP operation.

The objective functions of the Value Model and $H(\boldsymbol{\pi}_T)$ are:

$$\mathcal{L}_V = \mathbb{E}_i[(f_V^\phi(\mathcal{P}_i) - \hat{R}_i)^2] \tag{8}$$

$$H[\boldsymbol{\pi}_T] = -\mathbb{E}_i[\pi_{f_A^\theta}(A_T|\mathcal{P}_i)\log\pi_{f_A^\theta}(A_T|\mathcal{P}_i)] \tag{9}$$

where $f_V^\phi(\mathcal{P}_i)$ and $\hat{R}_i$ denote the Value Model's estimation of *State* $\mathcal{P}_i$ and the target value of real *Reward* $R_i$, respectively. $A_T$ denotes the specific action and $\pi_{f_A^\theta}(A_T|\mathcal{P}_i)$ is the probability that Policy $f_A^\theta$ takes action $A_T$ in state $\mathcal{P}_i$.

The detailed training procedure is shown in **Algorithm** 1.

The specific analysis of the time complexity of PKD training and testing are provided below:

The time complexity of PKD training is mainly divided into three parts: LLM fine-tuning (Line **2**), GNN-preference-driven Node Selector (Line **3-4**) and Node-preference-driven GNN Selector (Line **5-12**). The GNN-preference-driven Node Selector also can be divided into the annotations generation and prediction distillation.

Table 12: The GTA fine-tuning configurations on Llama-3.1-8B-Instruct.

| Model Name | Dataset Size | Epoch | lora_r | lora_alpha | Optimizer | Learning Rate | Time Cost |
|---|---|---|---|---|---|---|---|
| Llama-3.1-8B-Instruct | 53,617 | 2 | 4 | 4 | AdamW [57] | $1e^{-4}$ | 9h 41m 48s |

First, we use Low-Rank Adaptation (LoRA) strategy [58] for efficient parameter training, with hyperparameters set to $r = 4$, $\alpha = 4$, $epoch = 2$ (as shown in Table 12), and the rest are set according to the default settings of llama-factory[2]. Weight merge is also involved. In general, the time complexity of this part is $\mathcal{O}(nLdr + Ld^2r)$, where $n$ is the number of instructions, $L$ is the number of layers applying Lora, and d is the dimension of the LLM hidden layer. $r \ll d$, so the time complexity is bound by $\mathcal{O}(NLd + Ld^2)$.

Table 13: The time costs on **CORA** and **OGBN-ARXIV** of annotations generation by Llama-3.1-8B-Instruct.

| Dataset | CORA | OGBN-ARXIV |
|---|---|---|
| Time / GPU-hours | 0.11 | 1.68 |

The process of annotations generation includes sorting the selected nodes and the reasoning process of LLM, and its time complexity is $\mathcal{O}(W \log W)$ and $\mathcal{O}(WL'(l^2d + ld^2))$, where $W$ is the number of selected nodes, $L'$ is the number of transformer layers in LLM, and $l$ is the input sequence length. Generally, $W \ll l$, $L' \ll l$, then the time complexity is $\mathcal{O}(l^2d + ld^2)$.

Table 14: The time costs of retraining teacher GNNs on **CORA** and **OGBN-ARXIV**.

| Datasets | Teacher GNNs $(T_1, T_2, T_3, T_4)$ | Total running time (seconds) |
|---|---|---|
| **CORA** | GCN, GAT, APPNP, H2GCN | 2.4716 |
| **OBGN-ARXIV** | GCN, GAT, APPNP, H2GCN | 18.2017 |

The time complexity of teacher GNN (2-layers) re-training is bound by $\mathcal{O}((NF + M)D)$, where $N$ is the number of nodes, $F$ is the node feature dimension, $M$ is the number of edges, and $D$ is the GNN hidden layer dimension.

The time complexity of Node-preference-driven GNN Selector is $\mathcal{O}(W(l^2d + ld^2 + dd' + d'a))$, where $d'$ is the dimension of the MLP hidden layer, $a$ is the number of action categories, $W \ll l$, $a \ll d$, so the time complexity is bound by $\mathcal{O}(l^2d + ld^2 + dd')$. The training time complexity of student GNN is $\mathcal{O}((NF + M)D)$. Therefore, the overall time complexity is bound by $\mathcal{O}((L + 2l)d^2 + (d' + nL)d + 2l^2 + 2D(NF + M))$.

---

[2]https://github.com/hiyouga/LLaMA-Factory

Table 15: Peak memories and running times of Node-preference-driven GNN Selector on **CORA** and **OBGN-ARIXV**. "m" and "s" denote minute and second.

| Dataset | Peak memory | Running time / epoch |
|---|---|---|
| **CORA** | 454.62 MB | 7.9s |
| **OBGN-ARIXV** | 1655.18 MB | 44m 8.3s |

The inference time complexity of PKD is determined by the testing process of the student GNN. So its time complexity is bound by $\mathcal{O}((NF + M)D)$.

Specifically, We implement our proposed PKD with PyTorch (2.5.1) [59], PyTorch Geometric (2.6.1) [60], Python (3.10.16), Transformers (4.50.3), and vllm (0.7.0). We conduct all experiments on the NVIDIA A800-SXM4-80GB GPU and Intel(R) Xeon(R) CPU Max 9468.

The time costs for GTA fine-tuning, the Distance-based Neighbor Selector, LLM annotation, teacher re-training, and the PPO loop on Cora and Ogbn-Arixv are presented in Tables 12, 13, 14 and 15, respectively. The peak memories of performing PKD on the Cora and Ogbn-Arxiv are also listed in Table 15.

# D  Proofs for Propositions 3.1

The uncertainty usually refers to a measure of the confidence of a model in predicting a certain sample. From the perspective of collective consensus [40], we define the $K$-uncertainty of one node as the deviation of each teacher GNN's prediction probability distribution from the overall prediction probability distribution. From the **Proposition** 3.1, we can get that, the larger $\delta_K$ of one node, the stronger the uncertainty of this node, which is more beneficial to teacher GNNs training.

*Proof.* For each node $v$, the prediction probability distributions of $B$ teacher GNNs can be denoted by $P_1, P_2, ..., P_B$. The $K$-uncertainty of node $v$ is defined as:

$$\delta_K(v) \triangleq \sum_{1 \leq i < j \leq B}^{N} [D_{KL}(P_i(v)||P_j(v)) + D_{KL}(P_j(v)||P_i(v))] \tag{10}$$

Here, we define the average prediction probability distribution as following:

**Definition D.1.** The average prediction probability distribution $\mathcal{M}$ is the benchmark for the overall prediction probability distribution to measure both the models confidence and the consistency of each GNN with the overall probability distribution.

$$\mathcal{M}(v) = \frac{1}{B} \sum_{i=1}^{B} P_i(v) \tag{11}$$

Then, the uncertainty of node $v$ is,

$$\delta_v = \frac{1}{B} \sum_{i=1}^{B} D_{KL}(P_i(v)||\mathcal{M}(v)) \tag{12}$$

According to Jenson's inequality, we have

$$\delta_K(v) \geq N\delta_v \tag{13}$$

For any probability distribution $P$, we have

$$D_{KL}(P||\mathcal{M}) = H(P, \mathcal{M}) - H(P) \tag{14}$$

Then,

$$\delta_K = \frac{1}{N} \sum [H(P_i, \mathcal{M}) - H(P_i)] \tag{15}$$

As the $K$-uncertainty increases, the entropy of the GNN's prediction probability distribution $P_i$ increases, and the cross-entropy $H(P_i, \mathcal{M})$ grows significantly due to the larger probability distribution differences. So there is,

$$\delta_K(v) \propto \sum_{i=1}^{B} [H(P_i, \mathcal{M}) - H(P_i)] \propto \delta_v \tag{16}$$

That is,

$$\delta_K(v) \propto \delta_v \tag{17}$$

From the **Proposition** 3.1, we also can get that, selecting high-uncertainty nodes to expand the training set benefits GNNs training.

For a GNN with prediction probabilities $P(y = c|v)$, the entropy of an unlabeled node $v$ is

$$H(v) = -\sum_{c=1}^{C} P(y = c|v) \log P(y = c|v) \tag{18}$$

To maximize information gain, we select the node $v^*$ with the highest uncertainty (entropy):

$$v^* = \arg\max_{v} H(v) \tag{19}$$

After expanding $v^*$ to the training dataset, the loss function becomes:

$$\mathcal{L}_{new} = \mathcal{L}_{old}(\theta) + \mathcal{L}(f_\theta(v^*), y^*) \tag{20}$$

Here, $y^*$ is considered the true label based on the fine-tuned LLM. The GNN parameters are updated as:

$$\theta_{new} = \theta_{old} - r \cdot \nabla_\theta \mathcal{L}(f_\theta(v^*), y^*) \tag{21}$$

Since the prediction probability distribution of $v^*$ is close to uniform (due to high entropy) [61], the gradient more effectively corrects the GNN parameters [62], reducing the error. According to the preference rank: $\mathcal{V}_{\mathcal{PR}} = \text{Sort}(\{v_1, \ldots, v_N\}, \delta_K(v_1), \delta_K(v_2), \ldots, \delta_K(v_N))$, we can get the follows:

$$f_T^{\theta^*}(\tilde{\mathcal{D}}_L) = \arg\min_{v_i \in \{v_{\mathcal{PR}}^1, v_{\mathcal{PR}}^2, \ldots, v_{\mathcal{PR}}^W | \delta_K(v_{\mathcal{PR}}^W) > \tilde{\delta}_K\}} \frac{1}{W} \sum \mathcal{L}(f_T^\theta, v_i) \tag{22}$$

where $\tilde{\mathcal{D}}_L$ is the expanded training dataset. $f_T^{\theta^*}$ is the optimal parameter of teacher GNN. $v_{\mathcal{PR}}^w$ represents the $w$-th nodes in the preference rank.

$\square$

# E  Other Experimental Results

## E.1  Visualization

Figure 6 presents the outstanding node classification performance we mentioned in Section 4.2, which is illustrated by the t-SNE [63] visualization of the embedding spaces for CORA. Notably, Figure 6(a) illustrates the results of the student GNN (GCN) under the **# LN 5** condition.

From the Figure 6, we can see that, some KD methods fail to enable the student GNN to learn discriminative node representations, as evidenced by the absence of clustered structures in the embedding space, exemplified by MSKD, BGNN, and MTAAM. GCNII and KDGA struggle to form well-separated clusters, whereas methods like LLMGNN, GAugLLM, and FairGKD yield clusters with limited purity. Compared to these baselines, our method generates embeddings with significantly enhanced inter-class separability and high cluster purity, resulting in improved few-shot node classification performance.

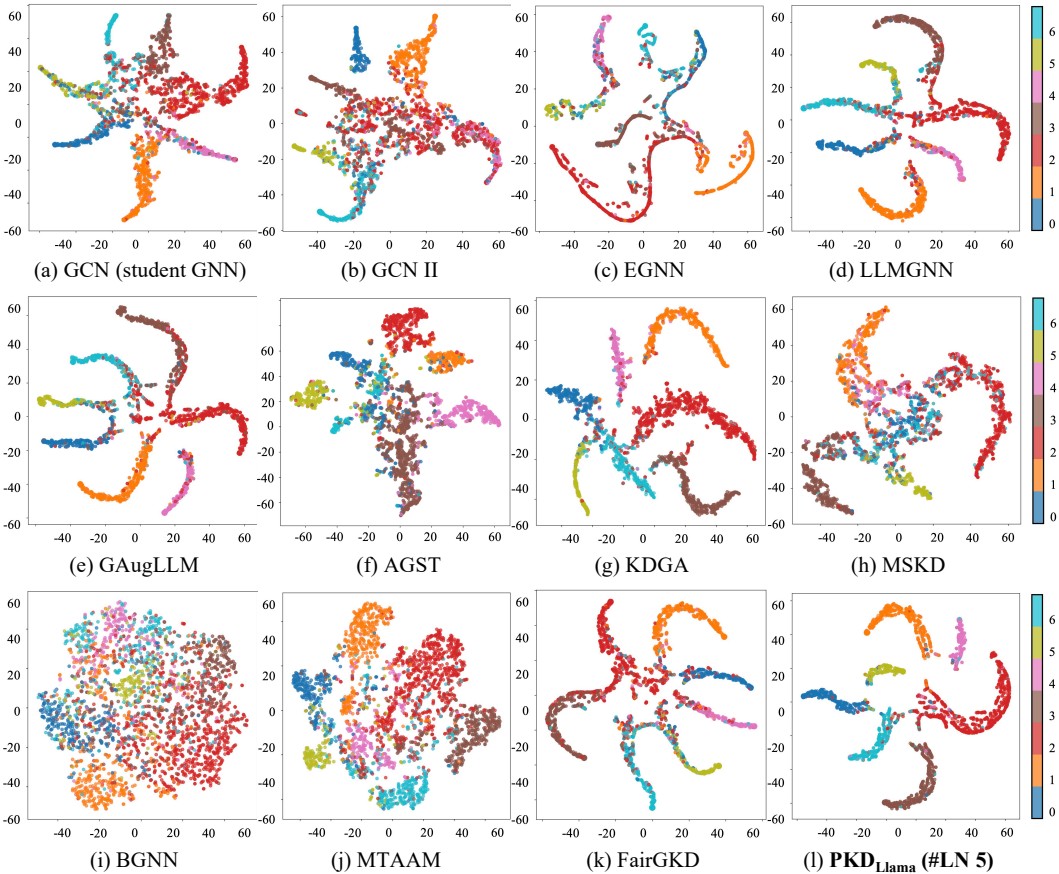

Figure 6: T-SNE [63] visualizations on CORA.

## E.2  Ablation Study

The results on Cora and Amazon Ratings using our PKD$_{Llama}$ (RL-based method) and the other three teacher selection methods (Entropy-based ranking, i.e., selecting the teacher GNN with the highest prediction confidence, Random selection, and End-to-end learning) are shown in Table 16. It is obvious that the RL-based method significantly outperforms the other three methods.

Table 16: Performance comparison of Entropy-based ranking, Random selection, End-to-end learning and RL-based approach on **CORA** and **AMAZON RATINGS**.

| Methods | CORA | AMAZON RATINGS |
|---|---|---|
| Entropy-based ranking | 75.70 | 55.74 |
| Random selection | 62.80 | 63.05 |
| End-to-end learning | 60.29 | 60.39 |
| $PKD_{Llama}$ (RL-based) | 90.27 | 65.93 |

## E.3 Hyperparameters Sensitivity Analysis

As mentioned in Sec. 4.4, we perform the hyperparameter sensitivity analysis over the loss-weight coefficients $\alpha, \beta, \gamma, \eta$ on two datasets, and report the results in Tables 17, 18, 19 and 20. As result, the proposed PKD can achieve much better performance when $\alpha = 0.5, \beta = 1, \gamma = 0.1, \eta = 0.3$.

Table 17: The influence of $\alpha$.

| $\alpha$ | 0.3 | 0.4 | 0.5 | 0.6 | 0.7 | 0.8 | 0.9 | 1 | 2 |
|---|---|---|---|---|---|---|---|---|---|
| CORA | 73.78 | 74.70 | 75.66 | 73.15 | 72.41 | 69.05 | 72.71 | 69.49 | 70.12 |
| AMAZON RATINGS | 61.81 | 62.86 | 63.83 | 62.76 | 61.30 | 60.28 | 60.47 | 61.66 | 61.74 |

Table 18: The influence of $\beta$.

| $\beta$ | 0.25 | 0.5 | 1 | 2 |
|---|---|---|---|---|
| CORA | 64.10 | 67.17 | 75.36 | 74.15 |
| AMAZON RATINGS | 62.94 | 63.23 | 63.91 | 63.93 |

Table 19: The influence of $\gamma$.

| $\gamma$ | 0.05 | 0.1 | 0.2 | 0.5 | 1 | 2 |
|---|---|---|---|---|---|---|
| CORA | 68.90 | 70.12 | 63.77 | 64.40 | 65.06 | 62.66 |
| AMAZON RATINGS | 62.58 | 64.01 | 63.71 | 63.63 | 63.93 | 63.93 |

Table 20: The influence of $\eta$.

| $\eta$ | 0.1 | 0.3 | 0.5 | 0.7 | 0.9 |
|---|---|---|---|---|---|
| CORA | 89.96 | 90.27 | 88.43 | 86.67 | 89.42 |
| AMAZON RATINGS | 64.87 | 65.93 | 64.79 | 63.75 | 63.81 |

## F Broader Impact

The proposed PKD offers significant broader impacts by enhancing few-shot node classification on TAGs. By combining the strengths of LLM and GNN, it improves learning efficiency, reducing the need for expensive and time-consuming manual annotation. This can benefit industries like social media, recommendation systems, and network analysis, enabling more accurate and scalable models for personalized services, fraud detection, and dynamic optimization.

Additionally, PKD can tailor message-passing mechanisms to node-specific attributes can lead to more adaptive and efficient machine learning models. It also democratizes access to advanced machine learning, allowing smaller organizations and researchers with limited resources to develop effective models. However, ethical considerations, such as privacy and fairness, must be prioritized to ensure responsible deployment.

