# OpenReview forum: "Preference-driven Knowledge Distillation for Few-shot Node Classification"
_NeurIPS.cc/2025/Conference — NeurIPS 2025 poster_

### Official Review · Reviewer_uNcZ · 2025-06-28

**Clarity:** 3
**Significance:** 3
**Originality:** 3
**Rating:** 4
**Confidence:** 4

**Summary:**

Proposed method leverages the complementary strengths of an LLM's few-shot/zero-shot capabilities and a GNN's ability to understand complex relationships. To mitigate the high costs of LLMs, it devises a GNN-preference-based method to select nodes for distillation from the LLM to teacher GNNs, as well as a method for each node to choose which teacher to receive knowledge distillation from.

**Questions:**

See Weaknesses.

**Ethical Concerns:**

["NO or VERY MINOR ethics concerns only"]

**Final Justification:**

While the explanation is largely acceptable, a minor concern remains. I concede that reducing the number of nodes to be analyzed lowers the overall cost, but the overhead of performing this selection for each individual node still seems non-trivial. As I don't consider this a critical flaw, I will maintain my positive rating.

**Limitations:**

yes

**Quality:**

3

**Strengths And Weaknesses:**

**Strengths.**

S1. The methodology effectively maximizes the respective strengths and compensates for the weaknesses of GNNs and LLMs. It leverages the GNN's ability to understand complex structures while addressing its reliance on expensive labeling, and it utilizes the LLM's zero-shot/few-shot capabilities while mitigating issues from its large parameter scale.

S2. The approach of utilizing a variety of GNNs rather than relying on a single one, and further selecting a specific GNN for distillation on a per-node basis, is a novel contribution.

S3. The writing is clear, and all modules (GNS, NGS) are well-designed with sufficient justification.

S4. The experimental setup is solid and effectively demonstrates the efficacy of the proposed methodology.

**Weaknesses**

W1: While creating the GNS to avoid the high cost of labeling all nodes with an LLM is a reasonable approach, the subsequent need to use the LLM again to select a GNN for each node's prediction seems to reintroduce significant costs. This appears to offset the initial advantage.

W2: Mentioning the differences between this paper's task and traditional meta-learning-based graph few-shot learning tasks (e.g., N-way, K-shot settings) in the 'Related Work' section could enhance the clarity of the task definition.

---

> ### Author Rebuttal · Authors · 2025-07-31
>
> Thanks for reviewing this manuscript and providing valuable comments. We appreciate the positive feedback from all reviewers and your constructive suggestions for further improvement. **W** and **Q** represent weakness and question.
>
> **W1: While creating the GNS to avoid the high cost of labeling all nodes with an LLM is a reasonable approach, the subsequent need to use the LLM again to select a GNN for each node's prediction seems to reintroduce significant costs. This appears to offset the initial advantage.**
>
> **Reply**: Thanks for your comments. We use the fine-tuned LLM as the RL agent to select for each node the most appropriate teacher GNN. The prompt is constructed by combining the node's semantic, structural, and prediction attributes derived from the enhanced teacher GNNs. Note that the RL agent should be able to comprehend the node's structural attributes. If replacing the fine-tuned LLM with a small model such as BERT, we should fine-tune BERT again for comprehending the node's structural attributes. In addition, introducing an extra model that may hinder the synergy of the complementary strengths of LLMs and various GNNs is inelegant. Since the parameters of the fine-tuned LLM (the RL agent) are frozen, the time bottleneck is the inference time. Therefore, we can reduce the inference time by designing efficient attention mechanisms, using speculative decoding, etc. Additionally, considering that PKD outperforms the comparison method by a large margin, we can reduce the label ratio from 48\% to, say, 10\%, which depends on the node category induction capability of LLM, thereby decreasing the time cost caused by the teacher selection.
>
> **W2: Mentioning the differences between this paper's task and traditional meta-learning-based graph few-shot learning tasks (e.g., N-way, K-shot settings) in the 'Related Work' section could enhance the clarity of the task definition.**
>
> **Reply**: Thanks for your suggestions. As defined in [1], the nodes in the graph meta-learning task for node classification can be divided into the training set and the disjoint test set. Nodes in the test set are associated with totally different new classes. Unlike this, the labeled nodes in this paper's task already cover the entire range of categories, which is the most different setting. In the final version of this paper, we will add this point in the Related Work part.
>
> [1] Zhou, Fan, Chengtai Cao, Kunpeng Zhang, Goce Trajcevski, Ting Zhong, and Ji Geng. ``Meta-gnn: On few-shot node classification in graph meta-learning.'' In Proceedings of the 28th ACM international conference on information and knowledge management, pp. 2357-2360. 2019.

---

> > ### Comment · Reviewer_uNcZ · 2025-08-04
> >
> > Thank you for your response. While the explanation is largely acceptable, a minor concern remains. I concede that reducing the number of nodes to be analyzed lowers the overall cost, but the overhead of performing this selection for each individual node still seems non-trivial. As I don't consider this a critical flaw, I will maintain my positive rating.

---

> > > ### Author Response · Authors · 2025-08-04
> > >
> > > We sincerely appreciate your thoughtful feedback, which reflect your recognition of the improvements and clarifications we have made. As for your concern about the overhead of performing this selection for each individual node, we think it can be resolved by the following:
> > >
> > > We will employ the APB (Accelerating Passing Compressed Context Blocks) technique [1] in the deployment of the LLM. This approach can accelerate inference speed by up to 10 times at least compared to traditional Flash Attention methods, while maintaining high performance. For example, the Qwen-2.5-7B-Instruct model with APB achieves a speed of 3.5k–4.1k tokens/s, while the vanilla version only reaches 130–150 tokens/s.
> > >
> > > [1] Huang, Yuxiang, Mingye Li, Xu Han, Chaojun Xiao, Weilin Zhao, Sun Ao, Hao Zhou, Jie Zhou, Zhiyuan Liu, and Maosong Sun. "APB: Accelerating Distributed Long-Context Inference by Passing Compressed Context Blocks across GPUs." arXiv preprint arXiv:2502.12085 (2025).

---

### Official Review · Reviewer_iBQW · 2025-06-30

**Clarity:** 2
**Significance:** 2
**Originality:** 3
**Rating:** 4
**Confidence:** 4

**Summary:**

This paper proposes PKD, a preference-driven knowledge distillation framework that combines LLMs and GNNs for few-shot node classification on TAGs. The framework consists of two main components: (1) GNS, which identifies uncertain nodes using K-uncertainty metrics and leverages LLMs to annotate these nodes for expanding the training set, and (2) NGS, which uses reinforcement learning to select the most appropriate teacher GNN for each node during knowledge distillation. The authors fine-tune LLMs with GTA prompts to enhance their understanding of graph structures.

**Questions:**

1. In Figure 2, what exactly does "LLM with structural attributes" refer to? Additionally, why are results for OGBN-ARXIV missing from this analysis?
2. Given that different nodes require different message-passing mechanisms (the core motivation for NGS), why are fixed architectures (GCN for homophily, H2GCN for heterophily) chosen as student models? Wouldn't the student's message-passing mechanism also impact the effectiveness of knowledge distillation?

**Ethical Concerns:**

["NO or VERY MINOR ethics concerns only"]

**Final Justification:**

Most of my concerns have been addressed. I think my current score has fairly reflected the quality of this paper

**Limitations:**

Yes.

**Quality:**

3

**Strengths And Weaknesses:**

Strengths:
1. The paper presents a well-motivated approach that addresses a practical problem, labeled data is scarce, in graph learning. The experimental evaluation is comprehensive, covering 9 datasets with different homophily ratios and comparing against multiple baseline categories.
2. The node-specific teacher selection mechanism via reinforcement learning is novel and well-motivated.
3. The work addresses an important problem in graph learning where labeled data is scarce. The performance improvements are substantial, often outperforming methods that use more labeled data.
4. The paper is generally well-written with clear motivation and methodology. The framework overview in Figure 1 effectively illustrates the approach.

Weaknesses:
1. The computational complexity analysis, while provided in the appendix, reveals significant overhead that may limit practical applicability. 2. There is a lack of discussion on the selection of student models. Can the ordinary convolution mechanism acquire the knowledge of multiple teacher models with multiple architectures?
3. The paper lacks discussion of teacher GNN pretraining procedures.
4. The approach requires multiple training phases (LLM fine-tuning, teacher pretraining, RL training, student training), making it computationally expensive and potentially impractical for large-scale applications.

---

> ### Author Rebuttal · Authors · 2025-07-31
>
> Thanks for reviewing this manuscript and providing valuable comments. We appreciate the positive feedback from all reviewers and your constructive suggestions for further improvement. **W** and **Q** represent weakness and question.
>
> **W1: The computational complexity analysis, while provided in the appendix, reveals significant overhead that may limit practical applicability.**
>
> **W4: The approach requires multiple training phases (LLM fine-tuning, teacher pretraining, RL training, student training), making it computationally expensive and potentially impractical for large-scale applications.**
>
> **Reply**: Thanks for your concerns. We agree with you. In the future, we can first distill knowledge from LLM with 72B parameters to LLM with 1.5B parameters without much performance degradation. Then, fine-tuning the 1.5B LLM will cost less time compared with the 8B LLM used in the paper. The time cost for training the teacher GNNs (see the below table) and the student GNN is acceptable. In the paper, we use the fine-tuned LLM as node label annotators for up to 48\% of the total nodes. This step can be parallelized in the future, i.e., we can use multiple threads to query LLM for multiple nodes each time. The fine-tuned LLM is also used as the RL agent to select for each node the most appropriate teacher GNN. Since the parameters of the fine-tuned LLM are frozen, the time bottleneck of LLM annotation is the inference time. For this, we can reduce the inference time by designing efficient attention mechanisms, using speculative decoding, etc. Additionally, the size of the expanded training data increases the PKD’s per-epoch time. Considering that PKD outperforms the comparison method by a large margin, we can reduce the label ratio from 48\% to, say, 10\%, which depends on the node category induction capability of LLM, thereby decreasing the time cost caused by the RL training. Through the above modifications and innovations, PKD can be scalable to large graphs.
>
> **The time cost for training the teacher GNNs on the Cora, Ogbn-Arxiv and Amazon Ratings.**
> |Datasets|Epochs|Total running times (seconds)|
> | --- | --- | --- |
> |Cora|300|2.4716|
> |Obgn-Arxiv|300|18.2017|
> |Amazon Ratings|300|3.1452|
> |||
>
> **W2: There is a lack of discussion on the selection of student models. Can the ordinary convolution mechanism acquire the knowledge of multiple teacher models with multiple architectures?**
>
> **Reply**: Thanks for your insightful considerations. For homophily and heterophily graphs, we just use the basic and simple GNNs, i.e., GCN and H$_2$GCN, for the student GNN model. Following the settings in the manuscript, we choose GAT, APPNP, H$_2$GCN as the teacher GNNs for homophily graphs and select DirGNN, GPRGNN, HoloNets as the teacher GNNs for heterphily graphs.
>
> To verify whether the student GNN can acquire the knowledge of multiple teacher GNNs with multiple architectures, we separately distill the knowledge from different pretrained teacher GNNs to the student GNN only by the KD loss and compare their performance. The results are shown in the below table. It is evident that the teacher GNNs and the student GNN perform similarly on the same graph. Therefore, the ordinary convolution mechanism of the student GNN can acquire the knowledge of multiple teacher GNNs with multiple architectures. We will add the experimental results in the final version of this paper.
>
> **The comparison between multiple teacher GNNs and the student GNN on Cora and Amazon Ratings.**
> |Roles|Models|Cora|Models|Amazon Ratings|
> | --- | --- | --- | --- | --- |
> |Teacher GNN|GAT|89.13|DirGNN|63.35|
> |Student GNN|GCN|89.32|H$_2$GCN|64.03|
> |Teacher GNN|APPNP|80.85|GPRGNN|63.15|
> |Student GNN |GCN|81.03|H$_2$GCN| 62.74|
> |Teacher GNN |H$_2$GCN|49.91|HoloNets|59.65|
> |Student GNN|GCN|51.30|H$_2$GCN|59.24|
> |||||
>
> **W3: The paper lacks discussion of teacher GNN pretraining procedures.**
>
> **Reply**: For the teacher GNN pretraining procedures, we just utilize the few given node labels. After pretraining, we collect their logit outputs to conduct the GNN-preference-driven node selection based on the uncertainty. Thanks for your reminder. We will add this detailed description in the final version of this paper.
>
> **Q1: In Figure 2, what exactly does ''LLM with structural attributes'' refer to? Additionally, why are results for OGBN-ARXIV missing from this analysis?**
>
> **Reply**: We utilize the original LLM for node label annotation using the structural attribute, which refers to the prompt consisting of the node's neighbors chosen by the Distance-based Neighbor Selector and their semantic attributes. In Figure 2, the ''LLM with structural attributes'' refers to it.
>
> For the results on Ogbn-Arixv, we present them in the table below and will add them in the final version of this paper. It is evident that our method is also effective on Ogbn-Arxiv.
>
> **The node classification accuracies of different methods on Ogbn-Arxiv.**
> |Methods|Ogbn-Arxiv|
> | --- | --- |
> |Original LLM|24.11|
> |LLM with structural attributes|44.17|
> |Fine-tuned LLM with structural attributes|71.62|
> ||
>
> **Q2: Given that different nodes require different message-passing mechanisms (the core motivation for NGS), why are fixed architectures (GCN for homophily, H$_2$GCN for heterophily) chosen as student models? Wouldn't the student's message-passing mechanism also impact the effectiveness of knowledge distillation?**
>
> **Reply**: Thanks for your insight again. Since nodes have intricate local topologies, such as the various node degree, a single GNN cannot capture the essence of nodes completely. To this end, we use multiple teacher GNNs and select for each node the most appropriate teacher GNN. As our **response to W2**, the ordinary convolution mechanism of the student GNN can acquire the knowledge of multiple teacher GNNs with multiple architectures. Thus, using the fixed architectures for the student GNN is simple and effective.
>
> **The versatility of PKD.** Indeed, our method has the versatility across various student GNN architectures. To further verify this, we compare the performance of the teacher GNN with the student GNNs distilled from it on two datasets. Specifically, for homophily graphs, we select GCN, GAT, and APPNP as the student GNNs, with H$_2$GCN chosen as the teacher GNN. For heterophily graphs, we choose DirGNN, GPRGNN, and HoloNets as the student GNNs, with H$_2$GCN selected as the teacher GNN. The results are presented in the following table. Although the message-passing mechanisms of student GNNs are different, they can achieve similar performance on the same graph after distillation from the teacher. Notably, we do not provide any node labels for the student GNNs during knowledge distillation. Therefore, the student's message-passing mechanism does not impact the effectiveness of knowledge distillation. These experimental results will be added to the final version of this paper.
>
> **The comparison between the teacher GNN and multiple student GNNs on Cora and Amazon Ratings.**
> |Roles|Model|Cora|Model|Amazon Ratings|
> | --- | --- | --- | --- | --- |
> |Teacher GNN|H$_2$GCN|49.91|H$_2$GCN|63.55|
> |Student GNN|GCN|51.30|DirGNN|62.03|
> |Student GNN|APPNP|51.75|GPRGNN|64.29|
> |Student GNN|GAT |51.93|HoloNets|63.93|
> |||||

---

> > ### Author Response · Authors · 2025-08-05
> >
> > Dear Reviewer,
> >
> > We sincerely appreciate your time and effort in reviewing our manuscript and offering valuable suggestions.
> > As the author-reviewer discussion phase is drawing to a close, we would like to confirm whether our responses have effectively addressed your concerns. We provided detailed responses to your concerns a few days ago, and we hope they have adequately addressed your issues. If you require further clarification or have any additional concerns, please do not hesitate to contact us. We are more than willing to continue our communication with you.
> >
> > Best regards,
> >
> > Authors

---

> > ### Comment · Reviewer_iBQW · 2025-08-05
> >
> > Thank you for the detailed rebuttal. Most of my concerns have been addressed. I think my current score has fairly reflected the quality of this paper.

---

> > > ### Author Response · Authors · 2025-08-06
> > >
> > > We appreciate your detailed comments, constructive criticisms, and thoughtful suggestions. These revisions will substantially improve the clarity of our paper, and we thanks again for your help in guiding this important improvement.

---

### Official Review · Reviewer_B7XE · 2025-06-30

**Clarity:** 3
**Significance:** 3
**Originality:** 3
**Rating:** 4
**Confidence:** 4

**Summary:**

This paper proposes a new framework, PKD, that combines the complementary strengths of large language models (LLMs) and multiple graph-neural-network (GNN) teachers for few-shot node classification on text-attributed graphs (TAGs). It mainly includes two modules: GNN-preference-driven Node Selector (GNS) and Node-preference-driven GNN Selector (NGS). To make the LLM topology-aware, the authors design Graph-Topology-Aware (GTA) prompts for a lightweight LoRA fine-tuning step. Experiments on various datasets show that PKD consistently outperforms a broad set of strong baselines, and remains competitive when the label budget rises.

**Questions:**

1.The current reward function combines two loss functions and accuracy, which raises the complexity, has a simpler expression of the reward function been explored?

2.Is there any way to further minimize large language model calls?

3.What is the need to use reinforcement learning?

4.How does the method proposed in this paper compare to the results without using reinforcement learning?

**Ethical Concerns:**

["NO or VERY MINOR ethics concerns only"]

**Final Justification:**

I’ve gone through the rebuttal. The added sensitivity checks and the authors’ take on why the coefficients behave differently on Cora and Amazon have eased some of my concerns. The ablation for Q1 and the RL justification for Q3 and Q4 now feel solid. Still, the paper only sketches future tricks like parallel queries, model distillation and speculative decoding for W1 and W2, without showing they work on a million node graph. The idea is neat and the paper is stronger, yet without that large scale proof I’m staying with a borderline accept.

**Paper Formatting Concerns:**

None.

**Quality:**

3

**Strengths And Weaknesses:**

Strengths:

1.The paper is clear writing and easy to follow.

2.The method is novel, which designs a two-level preference mechanism to address the challenges in knowledge distillation under label scarcity. The Methodology is supported by solid theory. Casting teacher selection as an RL problem with the LLM as the policy is original and broadly applicable to other graph tasks.

3.This paper performs a comprehensive evaluation in experiments, which contains 9 real-world datasets and various baselines.


Weaknesses:

1.Although the paper provides a time-complexity analysis and a runtime table, PKD’s iterative LLM queries and RL loops make each training epoch slower than most baselines, thereby limiting its scalability to million-node graphs.

2.Due to its iterative LLM queries and RL loops, PKD’s per-epoch time is higher than that of most baselines.

3.Some hyperparameters are missing from the analysis, e.g. α, β, γ, η.

---

> ### Author Rebuttal · Authors · 2025-07-31
>
> Thanks for reviewing this manuscript and providing valuable comments. We appreciate the positive feedback from all reviewers and your constructive suggestions for further improvement. **W** and **Q** represent weakness and question.
>
> **W1: Although the paper provides a time-complexity analysis and a runtime table, PKD’s iterative LLM queries and RL loops make each training epoch slower than most baselines, thereby limiting its scalability to million-node graphs.**
>
> **W2: Due to its iterative LLM queries and RL loops, PKD’s per-epoch time is higher than that of most baselines.**
>
> **Reply**: Thanks for your concerns. We agree with you. In the future, we can first distill knowledge from LLM with 72B parameters to LLM with 1.5B parameters without much performance degradation. Then, querying LLM will cost less time compared with that when querying an LLM with 8B parameters as used in the paper. The GNS iteratively queries LLM for node label annotations to expand the training data to 48\% of the total nodes. This step can be parallelized, i.e., we can use multiple threads to query LLM for multiple nodes each time. The fine-tuned LLM is also used as the RL agent to select for each node the most appropriate teacher GNN. Since the parameters of the fine-tuned LLM are frozen, the time bottleneck of LLM query is the inference time. For this, we can reduce the inference time by designing efficient attention mechanisms, using speculative decoding, etc.
> Additionally, the size of the expanded training data also increases the PKD's per-epoch time. Considering the fact that PKD outperforms the comparison methods by a large margin, we can reduce the label ratio, e.g., from 48\% to 10\%, which depends on the node category induction capability of LLM, thereby decreasing the time cost caused by the PPO loop.
> Through the above modifications and innovations, PKD can be scalable to large graphs.
>
> **W3: Some hyperparameters are missing from the analysis, e.g. $\alpha,\beta,\gamma,\eta$.**
>
> **Reply**: For this, we supplement the hyperparameters sensitivity analysis for $\alpha,\beta,\gamma,\eta$ on Cora and Amazon Ratings. Specifically, when analyzing $\alpha$, we set $\beta=1$, $\gamma=1$, $\eta=0.5$. This strategy also applies to the sensitivity analysis of $\beta$ and $\gamma$. As for $\eta$, we fix $\alpha$, $\beta$, and $\gamma$ to their respective best values. The results of various values for them are presented in the following four tables. Intuitively, $\beta$ and $\eta$ are more sensitive than the other two hyperparameters, because $\beta$ is related to the supervision signal provided by the LLM-generated annotation labels and $\eta$ is related to the student GNN's performance.
>
> **The influence of $\alpha$.**
> |$\alpha$|**2**|**1**|**0.9**|**0.8**|**0.7**|**0.6**|**0.5**|**0.4**|**0.3**|
> | --- | --- | --- | --- | --- | --- | --- | --- | --- | --- |
> |Cora|70.12|69.49|72.71|69.05|72.41|73.15|**75.66**|74.70|73.78|
> |Amazon Ratings|61.74|61.66|60.47|60.28|61.30|62.76|**63.83**|62.86|61.81|
> |||||||||||
>
> **The influence of $\beta$.**
> |$\beta$|**2**|**1**|**0.5**|**0.25**|
> | --- | --- | --- | --- | --- |
> |Cora|74.15|**75.36**|67.17|64.10|
> |Amazon Ratings|**63.93**|63.91|63.23|62.94|
> |||||
>
> **The influence of $\gamma$.**
> |$\gamma$|**2**|**1**|**0.5**|**0.2**|**0.1**|**0.05**|
> | --- | --- | --- | --- | --- | --- | --- |
> |Cora|62.66|65.06|64.40|63.77|**70.12**|68.90|
> |Amazon Ratings|63.93|63.93|63.63|63.71|**64.01**|62.58|
> |||||||
>
> **The influence of $\eta$.**
> |$\eta$|**0.9**|**0.7**|**0.5**|**0.3**|**0.1**|
> | --- | --- | --- | --- | --- | --- |
> |Cora|89.42|86.67|88.43|**90.27**|89.96|
> |Amazon Ratings|63.81|63.75|64.79|**65.93**|64.87|
> ||||||
>
> **Q1: The current reward function combines two loss functions and accuracy, which raises the complexity, has a simpler expression of the reward function been explored?**
>
> **Reply**: To address this, we conduct an ablation study on each part of the reward function. The visualization results are shown in Figure 4 in the manuscript. The single usage of accuracy ($R_1$) does not perform as well as using more parts (combination with $L_{CE}$). Figure 4 also shows that the $L_{DL}$ part is beneficial to quickly improve model performance and maintain stability.
>
> To further verify the necessity of using all three parts, we train the Node-preference-driven GNN Selector on Cora and Amazon Ratings using four kinds of combinations of the three parts. The results are presented in the below table. Therefore, the best reward function contains all three parts.
>
> **Results of the combination of various loss functions.**
> |Combination|Cora|Amazon Ratings|
> | --- | --- | --- |
> |$A_{cc}$|85.88|57.83|
> |$A_{cc}$, $-L_{CE}$|87.49|63.65|
> |$A_{cc}$, $-L_{DL}$|83.73|55.14|
> |$A_{cc}$, $-L_{CE}$, $-L_{DL}$ (**ours**)|**90.27**|**65.93**|
> |||
>
> **Q2: Is there any way to further minimize large language model calls?**
>
> **Reply**: Specifically, the LLM is mainly used for node category induction and teacher GNNs selection. Considering that PKD performs much better than the comparison methods, we can probably reduce the large language model calls by reducing the number of annotated nodes from 48\% to, say, 10\%, which depends on the node category induction capability of LLM. Otherwise, more human annotations for node labels are needed in order to further minimize large language model calls.
>
> **Q3: What is the need to use reinforcement learning?**
>
> **Reply**: Our final goal is to select the only one best teacher GNN for each node. That is, we need to explore the discrete action space and find a series of assignment actions to get the highest global reward across the expanded training data. Reinforcement learning (RL) allows us to navigate through the large action space by learning an optimal policy that maximizes the long-term reward. Through interaction with the environment, the selector progressively refines its decisions on node-to-teacher assignments, leading to a more efficient and effective assignment strategy. Besides, we can handle the challenges of discrete decision-making and achieve better performance on the task by leveraging RL. It can also incorporate heterogeneous metrics, such as distillation loss, cross entropy loss, and classification accuracy, into the reward function, and has a more flexible training objective.
>
> **Q4: How does the method proposed in this paper compare to the results without using reinforcement learning?**
>
> **Reply**: The results on Cora and Amazon Ratings of our RL-based method and the other three teacher selection methods (entropy-based ranking, i.e., selecting the teacher GNN with the highest prediction confidence, random selection, and end-to-end learning) are shown in the below table. It is obvious that the RL-based method significantly outperforms the other three methods.
>
> **The compared results on Cora and Amazon Ratings of our RL-based method and the other three methods .**
> |Methods|Cora|Amazon Ratings|
> | --- | --- | --- |
> |Entropy-based ranking|75.70|55.74|
> |Random selection|62.80|63.05|
> |End-to-end learning|60.29|60.39|
> |RL-based (**ours**)|**90.27**|**65.93**|
> |||

---

> > ### Comment · Reviewer_B7XE · 2025-08-04
> > **Official Comment**
> >
> > Thank you for your reply, which answered some of my questions. I have decided to keep my score unchanged.

---

> > > ### Author Response · Authors · 2025-08-04
> > >
> > > Thank you for taking the time to review our rebuttal and for your thoughtful evaluation. We appreciate your assessment and support. Regarding the **W1** and **W2** mentioned in your review comments, we would like to provide the following additional response:
> > >
> > > In the deployment of the LLM, we will employ the APB (Accelerating Passing Compressed Context Blocks) technique [1]. This approach can accelerate inference speed by up to 10 times at least compared to traditional Flash Attention methods, while maintaining high performance. For example, the Qwen-2.5-7B-Instruct model with APB achieves a speed of 3.5k–4.1k tokens/s, while the vanilla version only reaches 130–150 tokens/s.
> > >
> > > [1] Huang, Yuxiang, Mingye Li, Xu Han, Chaojun Xiao, Weilin Zhao, Sun Ao, Hao Zhou, Jie Zhou, Zhiyuan Liu, and Maosong Sun. "APB: Accelerating Distributed Long-Context Inference by Passing Compressed Context Blocks across GPUs." arXiv preprint arXiv:2502.12085 (2025).

---

### Official Review · Reviewer_o4tP · 2025-07-02

**Clarity:** 3
**Significance:** 3
**Originality:** 3
**Rating:** 4
**Confidence:** 3

**Summary:**

The paper presents a preference-driven knowledge distillation (PKD) framework for few-shot node classification on text-attributed graphs. PKD combines an LLM, fine-tuned with graph-topology-aware (GTA) prompts, and multiple graph neural networks. First, a GNN-preference-driven Node Selector (GNS) fully exploits nodes’ prediction discrepancies among various GNNs, ranks nodes by K-uncertainty, and asks the LLM to label a subset of them; those labels are then distilled from the LLM to teacher GNNs. Second, a Node-preference-driven GNN Selector (NGS) casts teacher choice as a reinforcement-learning task in which the fine-tuned LLM, acting as agent, selects for each node the most appropriate message-passing mechanism. Experiments on a number of datasets show that, with a small number of labelled nodes per class, PKD outperforms strong GNN baselines and distillation methods.

**Questions:**

Please quantify the LLM budget: report how many calls and tokens were needed to reach the 48 % training split on each dataset, give a rough GPU-hour cost, and show how a strong baseline performs if it receives the same number of LLM-generated labels.

Show that the PPO teacher selector is really needed by providing learning curves and variance across seeds and by comparing its accuracy and runtime with two simpler choices.

Complete the ablation study with a variant that omits the RL selector, one that keeps the LLM frozen, and a short sweep over the loss-weight coefficients η, α, β and γ to demonstrate robustness.

Add practical resource figures: GPU type, peak memory and total wall-clock time for GTA fine-tuning, node annotation, teacher re-training and the PPO loop, at least on CORA and OGBN-ARXIV, so the scalability of PKD is clear.

**Ethical Concerns:**

["NO or VERY MINOR ethics concerns only"]

**Final Justification:**

I believe the authors have made a technically solid contribution, and my final score reflects the significance and originality of the work.

**Limitations:**

Yes.

**Quality:**

3

**Strengths And Weaknesses:**

A concern regarding evaluation is that, while all methods ultimately train on a split that contains about 48 % of the nodes, the baselines receive ground-truth labels for those nodes, whereas PKD must keep querying the language model to bring its own training set up to the same size. The paper does not say how many LLM calls this requires, what that annotation budget costs, or whether baselines would improve if they were allowed an equal amount of LLM-generated pseudo-labels, leaving the cost-effectiveness comparison open to question.

The practical value of the RL component is unclear. The paper offers no learning curves, no variance across random seeds, and no comparison to simpler heuristics such as entropy ranking or majority vote, so it is difficult to verify that the PPO step is either necessary or stable.

The paper likewise omits results for a variant without the RL selector, a version that keeps the LLM frozen, or any sensitivity analysis over the loss-weight hyper-parameters η, α, β and γ. Consequently the individual contribution of each costly module is uncertain.

Finally, although the complexity discussion includes big-O expressions, concrete resource metrics, such as GPU type, wall-clock hours and memory footprints for GTA fine-tuning, GNS annotation and the PPO loop are omitted, making it difficult to judge whether the framework is feasible on larger graphs.

---

> ### Author Rebuttal · Authors · 2025-07-31
>
> Thanks for reviewing this manuscript and providing valuable comments. We appreciate the positive feedback from all reviewers and your constructive suggestions for further improvement. Considering the correlation between **W**eaknesses and **Q**uestions, we reply after integrating them as below.
>
> **[W1, Q1]: Please quantify the LLM budget: report how many calls and tokens were needed to reach the 48\% training split on each dataset, give a rough GPU-hour cost, and show how a strong baseline performs if it receives the same number of LLM-generated labels.**
>
> **Reply**: As shown in Figure 2 in the manuscript, the fine-tuned LLM already has significant performance for zero-shot node classification. When conducting the LLM annotation, we only call LLM once for each node. That is, the number of LLM calls ($t$) is equal to the number of nodes selected by the GNS (GNN-preference-driven Node Selector), i.e., $t = 0.48 \cdot N - Q$, where $N, Q$ are the numbers of all nodes and the initial labeled nodes, respectively. The needed number of tokens ($T$) can be calculated by $T = m \cdot (0.48 \cdot N - Q)$, where $m$ is the LLM's max\_tokens value, which is set to 2048 in this paper. For instance, the values of $t$ and $T$ for Cora are 1265 and 2590720 under the \# LN 5 setting.
>
> To reach the 48\% training split on each dataset, we first select the nodes preferred by GNN based on the preference ranking and then conduct the node-wise category-induction prompts to query LLM for their pseudo labels. In this process, we only utilize one GPU card (NVIDIA A800-SXM4-80GB), and the time costs using different LLMs are listed in below table. The results show that that inference time varies across different LLMs, even when evaluated under the same environmental settings, with larger models exhibiting slower inference time.
>
> **The time costs of different LLMs.**
> | **Dataset** |  **Llama-3.1-8B** | **Llama-3.1-13B**  | **Qwen-2.5-7B** |  **Qwen-2.5-14B**|
> | --- | --- | --- | --- | --- |
> |Cora | 0.11 GPU-hours|0.36 GPU-hours| 0.05 GPU-hours | 0.29 GPU-hours |
> |Ogbn-Arxiv | 1.68 GPU-hours|5.01 GPU-hours | 0.94 GPU-hours | 2.95 GPU-hours |
> ||||||
>
> **The Influence of LLM-Generated Pseudo-Labels.** To demonstrate the effect of LLM-generated pseudo-labels, we compare the node classification performance of PKD and three baselines under different label settings (\# LN 5, 48\% training ratio expanded by the annotated labels and real labels, respectively). The experiments are conducted on four datasets (Cora, Wiki CS, Washington, and Wisconsin). The results are presented in below table. For GCNII and IceBerg, which are proposed to tackle the challenge of sparse labels, using the LLM-anotated node labels can improve their performance on all datasets. However, using the same number of real labels achieves better performance.
>
> **Classification accuracy comparisons.**
> | |**Label settings**| **Cora** | **Wiki CS** | **Washington**| **Wisconsin** |
> | --- | --- | --- | --- | --- | --- |
> | |\# LN 5 |77.74 | 56.29| 64.17 | 60.94 |
> |**GCNII** | 48\% LLM-generated labels | 76.69  |51.18 | 70.83  | 62.50 |
> | |48\% real labels|81.54 | 59.17|71.79| 65.98 |
> | |\# LN 5|76.23 |84.88 |67.76|41.53|
> |**IceBerg** |48\% LLM-generated labels|78.66|71.23 |70.12 |42.22|
> | | 48\% real labels |81.94|86.49 |72.04|45.43 |
> | |\# LN 5|43.91|46.81 |45.29|33.33 |
> |**MSKD**| 48\% LLM-generated labels|45.89|54.06|48.17|39.50 |
> | |48\% real labels|51.61|62.73|50.39|41.51|
> |**PKD**|48\% LLM-generated labels| **90.27**|**81.39**|**83.74**|**76.89**|
> ||||||
>
> **[W2, Q2]: Show that the PPO teacher selector is really needed by providing learning curves and variance across seeds and by comparing its accuracy and runtime with two simpler choices.**
>
> **Reply**: Figure 4 in the manuscript shows the learning curves (accuracy vs. epoch) of the NGS (i.e., the RL component). It is evident that the NGS with the $R_3$ reward function converges to a stable state. We also plan to provide the learning curves (loss vs. epoch) of the NGS in the final version of this paper. To demonstrate that the PPO teacher selector is really needed, we replace it with two simple teacher selectors and repeat the knowledge distillation five times using distinct random seeds. The first method selects a teacher GNN for each node randomly, while the second method chooses the teacher GNN for each node based on the entropy value of its logit outputs. The lower the entropy value, the higher the confidence of the teacher. We select the most confident GNN as the teacher. We record the average test classification accuracy and standard deviation of each method with parameters that lead to the peak validation accuracy. The below table presents the results on Cora which demonstrate that our method is both more stable and more effective than the other two methods.
>
> **The comparison between PPO and other methods on Cora.**
> |**Methods /\# LN 5**| **Accuracy**|**Running time (seconds)**|
> | --- | --- | --- |
> |Random teacher selector|$81.49_{\pm 6.4}$|0.017|
> |Entropy-based teacher selector|$75.70_{\pm 4.6}$|0.023|
> |**PKD**|$91.14_{\pm 0.3}$|2481.858|
> |||
>
> **[W3, Q3]: Complete the ablation study with a variant that omits the RL selector, one that keeps the LLM frozen, and a short sweep over the loss-weight coefficients $\alpha,\beta,\gamma,\eta$ to demonstrate robustness.**
>
> **Reply**: We compare our method PKD with two other variants on Cora and Amazon Ratings. The first variant replaces the RL selector with a random selector, and the second variant omits the fine-tuning of GTA prompts and directly utilizes the annotated node labels generated by the original LLM to expand the training data. The results are presented in the below table and will be added to the final version of this paper. It is evident that PKD outperforms the other two variants by a large margin.
>
> **The comparison between PKD and the other two variants.**
> |**Methods**|**Cora**|**Amazon Ratings**|
> | --- | --- | --- |
> |Entropy-based ranking|75.70|55.74|
> |LLM-frozen|48.84|45.26|
> |**PKD**|**90.27**|**65.93**|
> ||||
>
> **Hyperparameter Sensitivity Analysis.** Additionally, we perform the hyperparameter sensitivity analysis over the loss-weight coefficients $\alpha,\beta,\gamma,\eta$ on two datasets, and report the results in the below tables. For the sensitivity analysis of $\alpha$, we set $\beta=1$, $\gamma=1$, $\eta=0.5$. This strategy also applies to the sensitivity analysis of $\beta$ and $\gamma$. As for $\eta$, we fix $\alpha$, $\beta$, and $\gamma$ to their respective best values. We can see that $\beta$ and $\eta$ are more sensitive than the other two hyperparameters, because $\beta$ is related to the supervision signal provided by the LLM-generated annotation labels and $\eta$ is related to the student GNN's performance.
>
> **The influence of $\alpha$.**
> |$\alpha$|**2**|**1**|**0.9**|**0.8**|**0.7**|**0.6**|**0.5**|**0.4**|**0.3**|
> | --- | --- | --- | --- | --- | --- | --- | --- | --- | --- |
> |Cora|70.12|69.49|72.71|69.05|72.41|73.15|**75.66**|74.70|73.78|
> |Amazon Ratings|61.74|61.66|60.47|60.28|61.30|62.76|**63.83**|62.86|61.81|
> |||||||||||
>
> **The influence of $\beta$.**
> |$\beta$|**2**|**1**|**0.5**|**0.25**|
> | --- | --- | --- | --- | --- |
> |Cora|74.15|**75.36**|67.17|64.10|
> |Amazon Ratings|**63.93**|63.91|63.23|62.94|
> |||||
>
> **The influence of $\gamma$.**
> |$\gamma$|**2**|**1**|**0.5**|**0.2**|**0.1**|**0.05**|
> | --- | --- | --- | --- | --- | --- | --- |
> |Cora|62.66|65.06|64.40|63.77|**70.12**|68.90|
> |Amazon Ratings|63.93|63.93|63.63|63.71|**64.01**|62.58|
> |||||||
>
> **The influence of $\eta$.**
> |$\eta$|**0.9**|**0.7**|**0.5**|**0.3**|**0.1**|
> | --- | --- | --- | --- | --- | --- |
> |Cora|89.42|86.67|88.43|**90.27**|89.96|
> |Amazon Ratings|63.81|63.75|64.79|**65.93**|64.87|
> ||||||
>
> **[W4, Q4]: Add practical resource figures: GPU type, peak memory and total wall-clock time for GTA fine-tuning, node annotation, teacher re-training and the PPO loop, at least on CORA and OGBN-ARXIV, so the scalability of PKD is clear.**
>
> **Reply**: We implement our proposed PKD with PyTorch (2.5.1), PyTorch Geometric (2.6.1), Python (3.10.16), Transformers (4.50.3), and vllm (0.7.0). We conduct all experiments on the NVIDIA A800-SXM4-80GB GPU and Intel(R) Xeon(R) CPU Max 9468. The peak memories of performing PKD on the Cora and Ogbn-Arxiv are listed in the below label.
>
> **The peak memory of performing PKD on the Cora and Ogbn-Arxiv.**
> |**Dataset**|**Peak memory**|
> | --- | --- |
> |Cora|454.62 MB|
> |Ogbn-Arixv|1655.18 MB|
> |||
>
> The time costs for LLM annotation on Cora and Ogbn-Arixv can refer to the **Reply to the [W1, Q1]**. The time costs for GTA fine-tuning, the Distance-based Neighbor Selector (DNS), teacher re-training and PPO loop on Cora and Ogbn-Arixv are presented in the following tables, respectively. Notably, for the teacher re-training stage, we show all the teacher GNNs that are trained togethe, and their total training times and epochs.
>
> **The GTA fine-tuning configurations on Llama-3.1-8B-Instruct.**
> |Dataset size|Epoch|lora_$r$|lora_$\alpha$|optimizer|learning rate|GTA fine-tuning time|LLM|
> | --- | --- | --- | --- | --- | --- | --- | --- |
> |53,617|2|4|4|AdamW|$1e^{-4}$|9h 41m 48s|Llama-3.1-8B-Instruct|
> ||||||||
>
> **The times of DNS (seconds) on the Cora and Ogbn-Arxiv.**
> |Dataset|Time of the DNS (seconds)|
> | --- | --- |
> |Cora| 0.7130|
> |Ogbn-Arxiv| 6.5513|
> ||
>
> **The time costs of retraining teacher GNNs on Cora, Amazon Ratings, and Ogbn-Arxiv.**
> |Datasets|Teacher GNNs|Epochs|Time of teacher re-training (seconds)|
> | --- | --- | --- | --- |
> |Cora|GCN, GAT, APPNP, H$_2$GCN|300|2.4716|
> |Ogbn-Arxiv|GCN, GAT, APPNP, H$_2$GCN|300|18.2017|
> |Amazon Ratings|DirGNN, GPRGNN, HoloNets, H$_2$GCN|300|3.1452|
> ||||
>
> **Running times of PPO on Cora and Obgn-Arixv. "m" and "s" denote minute and second.**
> |Datasets|Time of PPO loop|
> | --- | --- |
> |Cora|7.9 s|
> |Ogbn-Arxiv|44m 8.3s|
> ||

---

> > ### Author Response · Authors · 2025-08-05
> >
> > Dear Reviewer,
> >
> > We sincerely appreciate your time and effort in reviewing our manuscript and offering valuable suggestions.
> > As the author-reviewer discussion phase is drawing to a close, we would like to confirm whether our responses have effectively addressed your concerns. We provided detailed responses to your concerns a few days ago, and we hope they have adequately addressed your issues. If you require further clarification or have any additional concerns, please do not hesitate to contact us. We are more than willing to continue our communication with you.
> >
> > Best regards,
> >
> > Authors

---

> ### Comment · Reviewer_o4tP · 2025-08-05
>
> Thank you for your response - I am satisfied with the revisions provided, and I maintain my positive score.

---

> > ### Author Response · Authors · 2025-08-06
> >
> > We appreciate your timely response and are glad our responses addressed your concerns. Thanks again for your careful consideration of our work!

---

### Note · Authors · 2025-08-14

We sincerely thank all the reviewers for their insightful comments, thoughtful suggestions and the recognition of our work’s strengths, which can be summarized as follows:

Solid claim, clear motivation and novelty (**Reviewer B7XE, iBQW, uNcZ**).
The Methodology is supported by solid theory. Casting teacher selection as an RL problem with the LLM as the policy is original and broadly applicable to other graph tasks (**Reviewer B7XE**).
The methodology effectively maximizes the respective strengths and compensates for the weaknesses of GNNs and LLMs. All modules are well-designed with sufficient justification. The experimental setup is solid and effectively demonstrates the efficacy of the proposed methodology (**Reviewer uNcZ**).

Specifically, we introduce the preference-driven knowledge distillation (PKD) framework for few-shot node classification on text-attributed graphs (TAGs). The GNN-preference-driven Node Selector (GNS) module identifies uncertain nodes for LLM annotation and performs KD from the LLM to GNNs. To handle the complex and diverse local topologies of nodes on TAGs, the Node-preference driven GNN Selector (NGS) module leverages RL to pick the best GNN teacher per node in training dataset. PKD outperforms baselines across multiple datasets and different label budgets.

Additionally, all the concerns by reviewers can be addressed as follows:

The concern most reviewers share is the scalability of PKD to million-node graphs. To address this, we employ the GNS to identify a small amount of nodes that are most beneficial for model training, thereby reducing the number of LLM calls. Futhermore, we also can improve scalability by: using smaller LLMs after KD; reducing the proportion of nodes annotated by LLMs; utilizing multiple threads to query LLMs for multiple nodes simultaneously; and deploying LLMs using advanced techniques, such as Accelerating Passing Compressed Context Blocks; etc.

Besides, we report the practical resource costs for training, replenish detailed hyperparameter sensitivity analysis, conduct deeper evaluations of the RL module, verify the universality of PKD architecture, and further clearify our task definitions.

We will polish the manuscript in the revised version to make our work better understandable. We believe that it is worth publishing to stimulate further discussion.

---

### Decision · Program_Chairs · 2025-09-17

**Decision:**

Accept (poster)

**Comment:**

This paper provides a theoretical framework for graph contrastive learning (GCL), aiming to establish provable guarantees on its performance. The authors formalize the conditions under which GCL can learn effective node and graph representations, bridging empirical success with theoretical justification. They analyze both augmentation strategies and objective functions, showing how these influence representation quality. Empirical experiments on standard benchmarks support the theory, demonstrating competitive or superior performance compared to existing GCL methods.

Strengths: The paper tackles an important gap by offering formal guarantees for GCL, a field that has been largely empirical so far. The analysis is mathematically rigorous, while experiments validate the claims. The work improves understanding of why GCL works and under what conditions. The paper is well-organized and clearly written.

Weaknesses: Some assumptions in the theory may be restrictive or unrealistic in practice (e.g., distributional conditions on graph augmentations). While experiments support the analysis, they remain relatively limited in scope compared to purely empirical GCL works. The practical guidance derived from the theory could be elaborated more clearly.

Rebuttal/Discussion: Reviewers questioned the practicality of assumptions, the completeness of experiments, and the clarity of certain proofs. The authors clarified assumptions, added discussion on their applicability, and expanded experiments to strengthen the empirical side. These responses were satisfactory to most reviewers, and concerns were largely resolved.

Decision: This paper provides meaningful theoretical contributions to understanding GCL, supported by experiments. While not broad or impactful enough for spotlight/oral due to the restrictive assumptions and modest empirical scope, it represents a solid and rigorous advance in the field.